# A gene regulatory network controls the balance between mesendoderm and ectoderm at pluripotency exit

Hanna L Sladitschek[†] & Pierre A Neveu[*] (iD)

## Abstract

During embryogenesis, differentiation of pluripotent cells into somatic cell types depends both on signaling cues and intrinsic gene expression programs. While the molecular underpinnings of pluripotency are well mapped, much less is known on how mouse embryonic stem cells (mESCs) differentiate. Using RNA-Seq profiling during specification to the three germ layers, we showed that mESCs switched on condition-specific gene expression programs from the onset of the differentiation procedure and that primed pluripotency did not constitute an obligatory intermediate state. After inferring the gene network controlling mESC differentiation, we tested the role of the highly connected nodes by deleting them in a triple knock-in *Sox1-Brachyury-Eomes* mESC line reporting on ectoderm, mesoderm, and endoderm fates. This led to the identification of regulators of mESC differentiation that acted at several levels: Sp1 as a global break on differentiation, Nr5a2 controlling ectoderm specification, and notably Fos:Jun and Zfp354c as opposite switches between ectoderm and mesendoderm fate.

**Keywords** differentiation; germ layer; embryonic stem cells; epiblast stem cells; pluripotency
**Subject Categories** Chromatin, Transcription & Genomics; Stem Cells & Regenerative Medicine
**Mol Syst Biol. (2019) 15: e9043**

## Introduction

During development, the differentiation of the initial pool of pluripotent cells into a great variety of somatic cell types is thought to depend on signaling cues and intrinsic gene expression programs. Embryonic stem cells (ESCs) are an ideal model system to study this phenomenon as they can be maintained as continuous cell lines that possess the dual ability to self-renew and to differentiate into any somatic cell fate found in the adult organism depending on the applied culture regime (Nichols & Smith, 2009). Indeed, optimized

*in vitro* differentiation protocols have been devised that guide mouse ESCs (mESCs) to acquire fates of the three primary germ layers—ectoderm (Ying *et al*, 2003), mesoderm (Torres *et al*, 2012), and endoderm (Borowiak *et al*, 2009). Such *in vitro* procedures allow to precisely delineate the hierarchy and dynamics of gene expression changes in response to a defined, homogeneous, and constant external signaling environment. The current paradigm of fate acquisition from an mESC state is a transition from naïve pluripotency to primed pluripotency to differentiated cells (Smith, 2017). However, the interrelationship between different commitment programs is poorly characterized as the vast majority of studies focus on a single fate decision (Ying *et al*, 2003; Keller, 2005; Borowiak *et al*, 2009; Torres *et al*, 2012). Moreover, markers that are specific for the desired fate are used and the potential existence of other fates among the cell population is generally not addressed. While much attention has been devoted to the gene regulatory networks underlying pluripotency (Loh *et al*, 2006; Chen *et al*, 2008; Kim *et al*, 2008; Dunn *et al*, 2014) or reprogramming (Dunn *et al*, 2019), the networks governing mESC differentiation are largely unexplored. The lack of data for intermediate differentiation stages further complicates the delineation of fate decisions as gene expression trajectories.

Here, we take a integrated systems approach to investigate germ layer specification from mESCs. Conducting mRNA sequencing at high temporal resolution revealed that gene expression programs diversified in a germ layer-specific manner from the onset of differentiation, with primed pluripotency only being an intermediate state of endodermal differentiation. We inferred the gene regulatory network governing mESC differentiation, identifying a small number of highly connected nodes as potential novel regulators of differentiation. We combined a triple knock-in *Sox1-Brachyury-Eomes* mESC line reporting simultaneously on the acquisition of ectoderm, endoderm, and mesoderm and CRISP/Cas9-mediated knockout to test the functionality of the highly connected nodes. We showed that these can have three main functions: (i) general regulation of differentiation like for Sp1, (ii) control of specific fates like Nr5a2 for ectoderm specification, and (iii) switch between fates. As representatives of the last category, Fos:Jun biased mESC differentiation toward ectoderm at the expense of endoderm while Zfp354c had

European Molecular Biology Laboratory, Cell Biology and Biophysics Unit, Heidelberg, Germany
*Corresponding author. Tel: +49 6221 387 8336; E-mail: neveu@embl.de
†Present address: Department of Molecular Medicine, University of Padua School of Medicine, Padua, Italy

the reverse effect. Thus, our strategy to predict gene regulatory networks followed by the development of multicolor fluorescent reporter lines and interference with CRISP/Cas9 to quantitatively test the involvement of nodes is particularly adapted to find novel regulators of mESC differentiation.

# Results

## Common gene expression changes during mESC differentiation to the three germ layers

We reasoned that profiling gene expression at sufficient temporal resolution would establish the relatedness of gene expression changes between different fate acquisitions. We therefore differentiated mESCs toward precursors of the three primary germ layers using established protocols reported in the literature (Ying *et al*, 2003; Borowiak *et al*, 2009; Torres *et al*, 2012). At the end of the procedure (after 6 days of differentiation), cultures differentiated to ectoderm, mesoderm, or endoderm stained positive for their respective fate markers: TUJ1, a neuronal marker; DESMIN, a marker of muscle cells; or GATA6, an endoderm marker (Fig 1A). We next sampled gene expression at high temporal resolution for each of these three regimes (Fig 1B), in order to accurately capture the timing of gene expression changes and to identify genes that are subject to transient up or down-regulation.

In the three distinct differentiation protocols, we found a similar number of genes exhibiting more than fourfold expression changes (5,519 genes fulfilled this criterion for neuroectodermal differentiation, 5,418 genes for mesodermal, and 4,730 genes for endodermal differentiation, Fig 1C). The majority of these differentially regulated genes displayed an increase of expression during differentiation (Fig 1D–F). Surprisingly, 3,370 genes—representing roughly two-thirds of the set of differentially regulated genes—were common among the three different fate commitments, suggesting a role in the exit from pluripotency or the silencing of self-renewal rather than an involvement in specific cell fate decisions. This list contained the *bona fide* pluripotency markers *Oct4*, *Nanog*, and *Rex1* (or *Zfp42*) that are also known as potent regulators of a pluripotent cell identity.

In order to determine the broad biological functions that were most affected during differentiation, we performed enrichment analysis (Ashburner *et al*, 2000) of the curated KEGG (Kyoto Encyclopedia of Genes and Genomes) pathways (Kanehisa *et al*, 2016). This highlighted genes associated with extracellular matrix–receptor interactions and focal adhesions as a common signature of differentiation (Appendix Fig S1A and B). Indeed, we observed a potent upregulation of both integrins and different collagen types in all three differentiation regimes (Appendix Fig S1C). Interestingly, genes associated with epithelial–mesenchymal transition (EMT), such as *Twist* and *Slug*, were upregulated (Appendix Fig S1D). However, E-cadherin (*Cdh1*) transcript levels were only weakly downregulated, while mRNA levels of N-cadherin (*Cdh2*) and Vimentin (*Vim*) increased 10-fold or more (Appendix Fig S1E). KEGG analysis also identified cytokine–cytokine receptor interactions as a functional layer strongly affected by the rewiring of gene expression in cells undergoing differentiation (Appendix Fig S2). Collectively, these findings reflect the

required adaption to a different repertoire of cytokines and the remodeling of cell–cell and cell–matrix interactions occurring during cellular differentiation.

## Early divergence of fate-specific gene expression programs

Next, average linkage clustering was performed to gauge the relatedness of gene expression profiles in cells undergoing differentiation (Fig 2A). Undifferentiated mESCs and adult mouse tissues were used as reference profiles. Already from day 1 onwards, the transcriptome of cells differentiating toward an ectodermal fate clustered with the transcriptome of differentiated tissues. In contrast, cell populations undergoing endoderm or mesoderm differentiation clustered with undifferentiated mESCs for the first few days. From day 4 of differentiation procedure onwards, the transcriptomes of cells undergoing mesoderm and endoderm differentiation clustered with differentiated tissues. Interestingly, this time point coincided with splitting of the mesendoderm cluster into two separate clusters distinguishing a mesodermal and an endodermal fate.

We used principal component analysis (PCA) to visualize the trajectories of gene expression signatures of cells undergoing differentiation and capture the differences between them (Fig 2B). The first principal component (PC) PC1 representing 44.3% of the variation was contributed by genes expressed in the pluripotent state as well as genes upregulated in the three differentiation regimes (Fig 2C). The second and third PCs, PC2, and PC3 representing 20.5 and 8.7% of the variation could discriminate the three differentiation trajectories from day 1 (Fig 2D). These findings demonstrate that distinct differentiation cues instruct a pluripotent population to immediately start to implement fate-specific gene expression programs. This immediate divergence of gene expression profiles suggests that culture-induced differentiation does not proceed via common states with a gradually restricted pluripotent potential.

We compared germ layer specification trajectories obtained from *in vitro* mESC differentiation with published transcriptomes originating from spatially defined regions of gastrulating mouse embryos (Peng *et al*, 2019). Transcriptomes of sections from E5.5 and E6.0 embryos projected at the beginning of the differentiation trajectories in a similar location as the transcriptomes of mESCs will low Nanog expression (Fig EV1A–C). Expression profiles of proximal posterior sections from E6.5 onwards (corresponding to the location of the primitive streak in the embryo) projected on the *in vitro* endoderm differentiation trajectory (Fig EV1D–F), in accordance with the definitive endoderm originating from the primitive streak (Lewis & Tam, 2006). Transcriptomes of proximal mesoderm sections at E7.0 (Fig EV1E) projected on the *in vitro* mesoderm differentiation trajectory. Finally, the expression profiles of some sections of the anterior epiblast at E7.0 and E7.5 projected on the *in vitro* ectoderm differentiation trajectory (Fig EV1E and F), the anterior epiblast giving rise to ectoderm in mouse embryos (Tam & Behringer, 1997). Notably, the specification of different regions of the mouse epiblast from E6.5 onwards was asynchronous as some sections retained a more undifferentiated character as revealed by projection on our PC1–PC2 map (Fig EV1D–F). Thus, *in vitro* differentiation to endoderm, mesoderm, and ectoderm recapitulated *in vivo* germ layer specification: (i) the *in vitro* endoderm differentiation resembling primitive streak formation, (ii) the *in vitro* mesoderm differentiation resembling proximal

                    

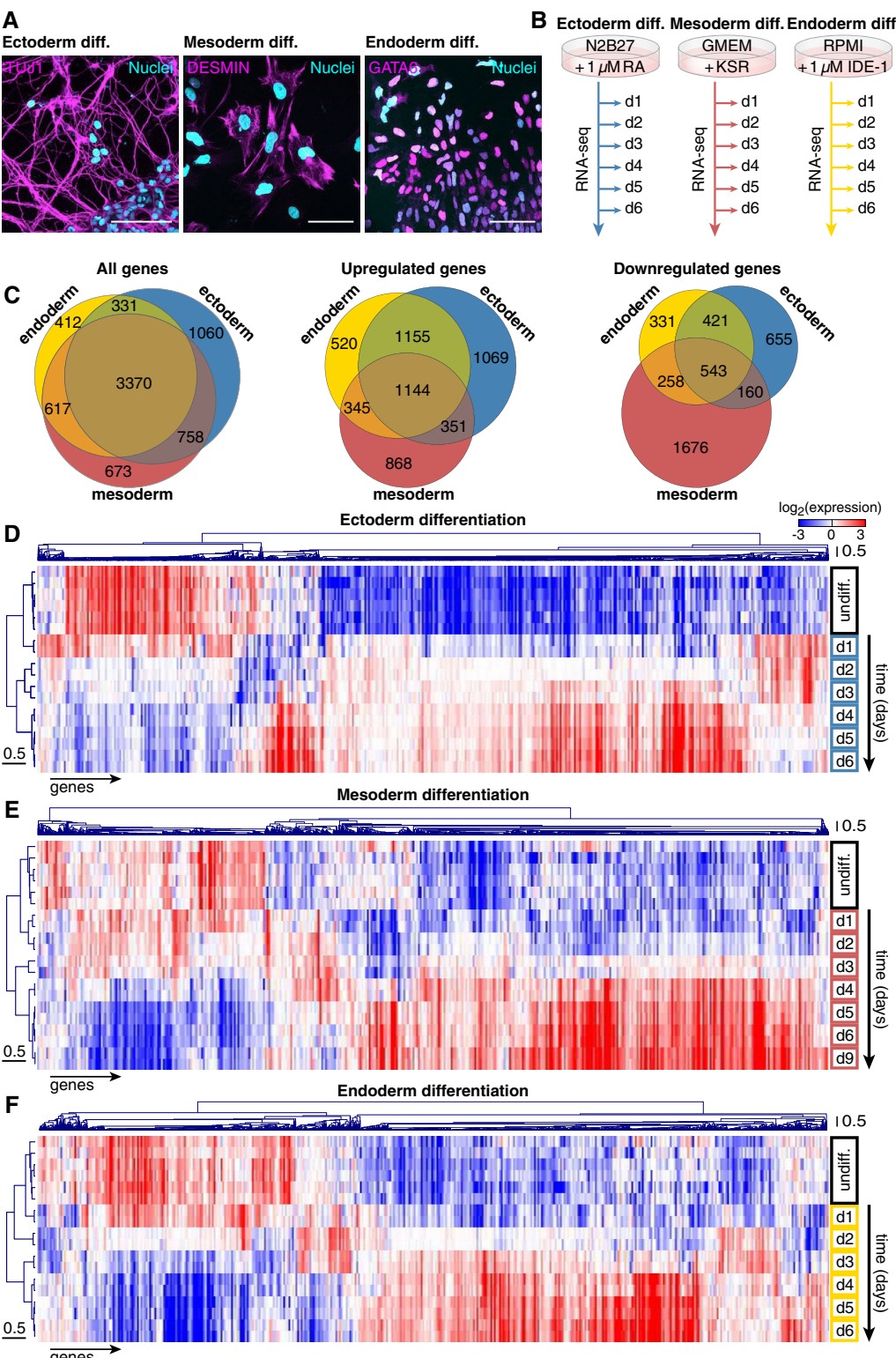

**Figure 1. Comparison of gene expression changes during mESC differentiation toward the three germ layers.**

A    Immunostaining of TUJ1, DESMIN, and GATA6 after 6 days of mESC differentiation toward ectoderm, mesoderm, and endoderm. Scale bar: 50 μm.

B    Scheme of the experimental approach to quantitatively capture gene expression changes during mESC differentiation toward the three germ layers at high temporal resolution.

C    Venn diagrams of genes with differential expression during mESC differentiation toward the three germ layers.

D–F  Hierarchical clustering of mRNA expression of mESCs differentiated toward ectoderm (D), mesoderm (E), and endoderm (F).

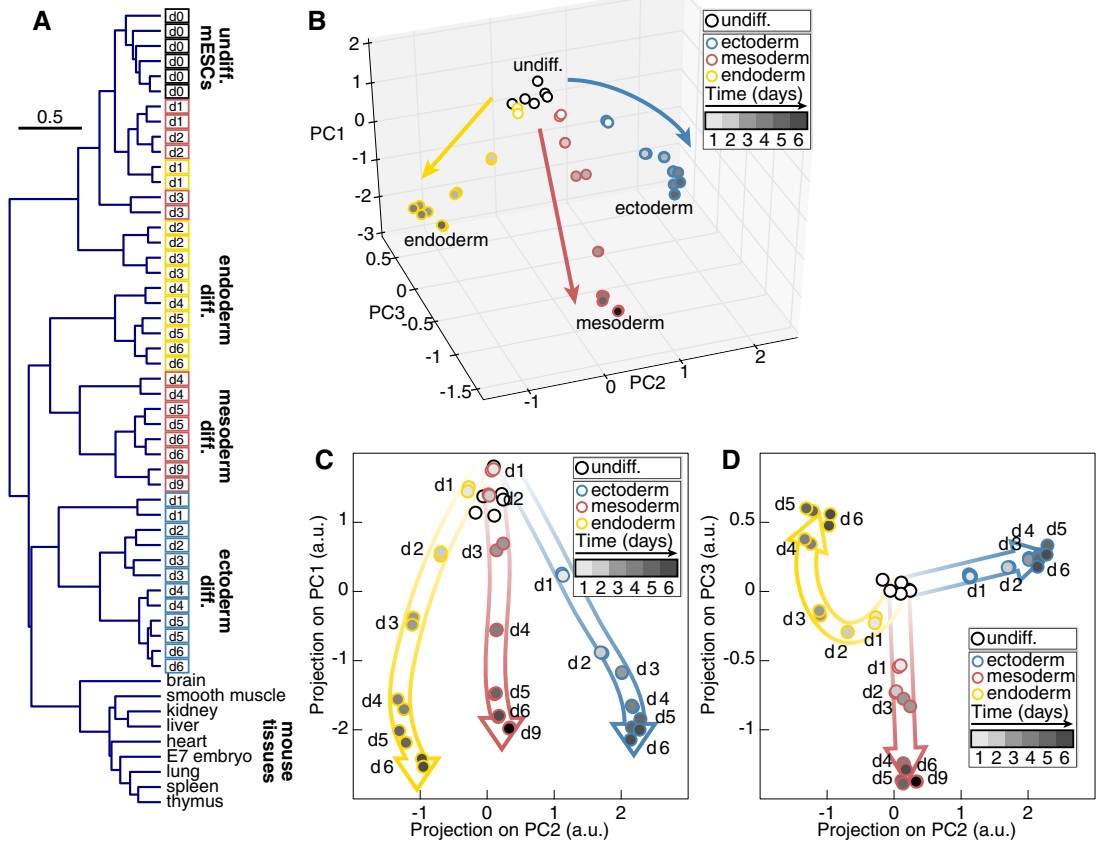

**Figure 2. Early divergence of fate-specific gene expression programs.**

A    Hierarchical clustering of gene expression profiles of mESCs differentiated toward the three germ layers and differentiated mouse tissues.

B    Principal component analysis of gene expression changes during mESC differentiation.

C, D    Projection on the principal components PC1, PC2 (C) and PC2, PC3 (D) of gene expression profiles during mESC differentiation.

embryonic mesoderm, and (iii) the *in vitro* ectoderm differentiation resembling the ectoderm specification from the anterior epiblast (Peng *et al*, 2019).

**Primed pluripotency represented by epiblast stem cells is not a common differentiation intermediate *in vitro***

We went on to investigate how our "fate map" relates to the notion of pluripotent cells exiting naïve pluripotency via a state of "primed" pluripotency prior to engaging in specific cell fate decision making (Kalkan & Smith, 2014). Epiblast stem cells (EpiSCs) are an established model of a population with a restricted (or primed) pluripotent potential and indeed thought to represent the *in vitro* equivalent of the mouse postimplantation epiblast (Brons *et al*, 2007; Tesar *et al*, 2007; Nichols & Smith, 2009). EpiSCs can be derived from mESCs using a chemically defined medium containing Fgf2 and Activin A (Guo *et al*, 2009). We followed the changes in the gene expression signature during this interconversion from naïve to primed pluripotency (Fig 3A). Projection of the sampled gene expression profiles onto the PC1–PC2 map revealed that EpiSCs specification followed the trajectory of endodermal differentiation (Fig 3B). Interestingly, EpiSCs did not progress all

the way along this trajectory but remained stabilized in an intermediate state of differentiation (Fig 3B and C). Closer inspection of known marker genes confirmed that mature pluripotent EpiSCs closely resembled an intermediate state of endodermal differentiation with high Oct4 expression but low Sox2 and Rex1 levels (Fig EV2A). Moreover, endoderm fate markers were upregulated to similar levels in both mature pluripotent EpiSCs and endodermal precursors (Fig EV2B). In fact, both *in vitro*-differentiated EpiSCs (Fig EV2C) and published embryo-derived EpiSCs (Tesar *et al*, 2007) (Fig EV2D) had downregulated expression of naïve pluripotency markers and upregulation of markers of primed pluripotency (Tesar *et al*, 2007) compared to mESCs. Finally, expression profiles of embryo-derived EpiSCs (Tesar *et al*, 2007) projected in the same location of the PC1–PC2–PC3 map as the profiles of *in vitro*-differentiated EpiSCs (Fig EV2E and F). The position of EpiSCs in the landscape of gene expression profiles prompted us to ask where mESCs with low Nanog expression or mESCs cultured in "ground state" pluripotency conditions (also known as "2i") (Ying *et al*, 2008) would reside in that landscape. Transcriptomes of mESCs with low Nanog levels projected on the first 2–3 days of the mesoderm and endoderm differentiation trajectories (Fig EV2G and H) but were markedly distinct from

EpiSC expression profiles, notably by the absence of expression of endoderm markers. In contrast, mESCs maintained in "2i" resembled undifferentiated mESCs grown in "LIF+serum" or early mesoderm and ectoderm differentiation intermediates (Fig EV2G and H). Altogether, our results support the notion that *in vitro* differentiation proceeds only for endoderm differentiation via an EpiSC-like state of primed pluripotency. Notably, gene expression trajectories for ectodermal and mesodermal differentiation appear to be preconfigured toward their prospective fate straight from the exit from naïve pluripotency. Therefore, primed pluripotency does not constitute an intermediate state of mesodermal or ectodermal differentiation *in vitro*.

## A general transcriptional network governing mESC differentiation

In order to identify the gene regulatory network that would regulate this common differentiation program, we predicted binding sites for transcription factors with curated weight matrices (Mathelier *et al*, 2016) in the upstream 1 kb proximal region of promoters. Motif activities were computed using an additive model of motif contribution to gene expression (Bussemaker *et al*, 2001). Genes with significant motif activities were then considered to build a transcriptional network regulating mESC differentiation (Fig 4A). Surprisingly, pluripotency factors were not present in this network. However, the Nanog motif activity increased in the first days of differentiation and was anti-correlated with Nanog mRNA levels ($r = -0.68$, Fig EV3A and B), consistent with Nanog being associated with transcriptional repression complexes (Liang *et al*, 2008). This network was much less connected and almost devoid of feedback loops compared to the core pluripotency network (Chen *et al*, 2008; Kim *et al*, 2008).

Instead, the network made extensive use of feed-forward loops, a hallmark of transcriptional networks (Milo *et al*, 2002) and dense overlapping regulon (DOR) motifs. The large number of DOR suggests that cells integrate multiple inputs during cell fate acquisition. The highly connected nodes Meis3, Sp1, Gabpa, Nr5a2, Foxj2, Fos:Jun, and Atf1 were input nodes in DOR motifs. In such motifs, Fos:Jun exerted interactions of opposite sign compared to the other input nodes Sp1, Nr5a2, Gabpa, and Atf1. However, the behavior of DOR motifs cannot be predicted from their topology alone (Alon, 2007).

While we found that *in vitro* differentiation protocols allow to bypass primed pluripotency, EpiSCs are themselves pluripotent. In theory, critical components of the transcriptional network could be reused during germ layer specification starting from a state of primed pluripotency. We thus set out to determine the transcriptional network underlying fate specification from EpiSCs. We measured by deep sequencing gene expression at high temporal resolution during EpiSC differentiation to the three germ layers (Fig 4B). Projecting the gene expression profiles on the previously computed PCs, we observed that the three trajectories showed progression along PC1, the general differentiation axis, but could not be distinguished by PC2 (Fig EV3C). Nonetheless, PCA conducted on EpiSC differentiation data alone determined that germ layer specification from EpiSCs followed distinct trajectories that diverged within 1 or 2 days after onset of differentiation cues (Fig EV3D). This suggested that EpiSC differentiation shared some similarities with mESC differentiation. Indeed, we found that a much simpler gene regulatory network underlay EpiSC differentiation, sharing with the mESC differentiation network 35 out of 49 nodes, among them all the highly connected nodes (Fig 4C). In fact, the vast majority of the nodes unique to the EpiSC differentiation

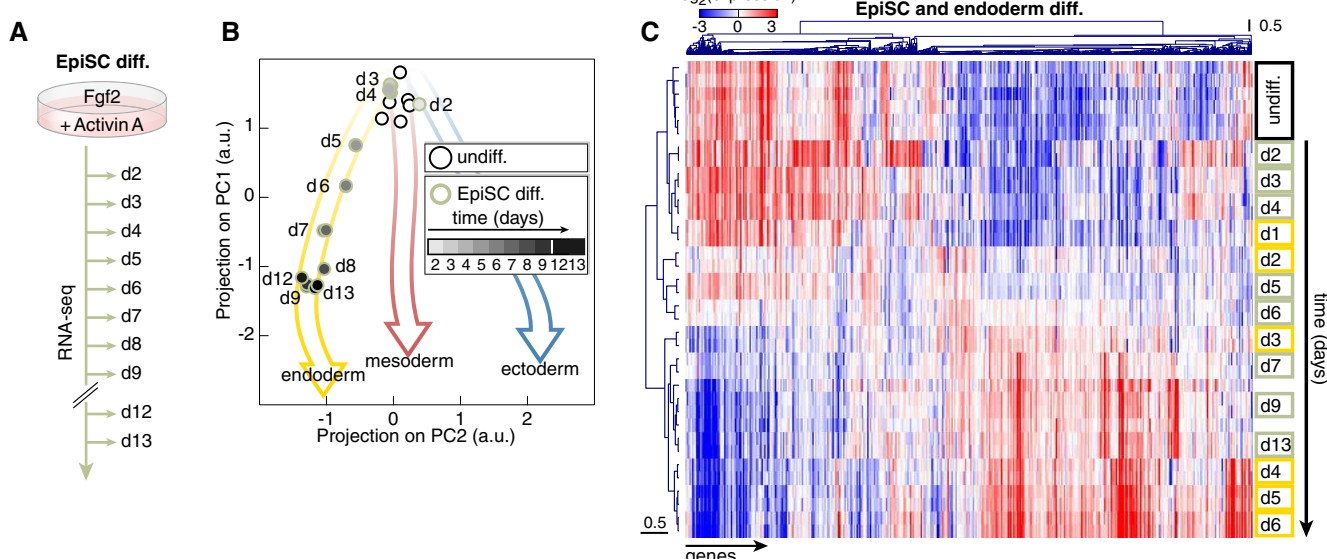

**Figure 3. Primed pluripotency is not a common differentiation intermediate *in vitro*.**

A    Scheme of the experimental approach to quantitatively capture gene expression changes during mESC differentiation toward EpiSCs.

B    Projection on PC1 and PC2 of gene expression profiles of mESCs differentiated toward EpiSCs.

C    Hierarchical clustering of mRNA expression during differentiation of mESCs toward endoderm (yellow) or EpiSCs (green).

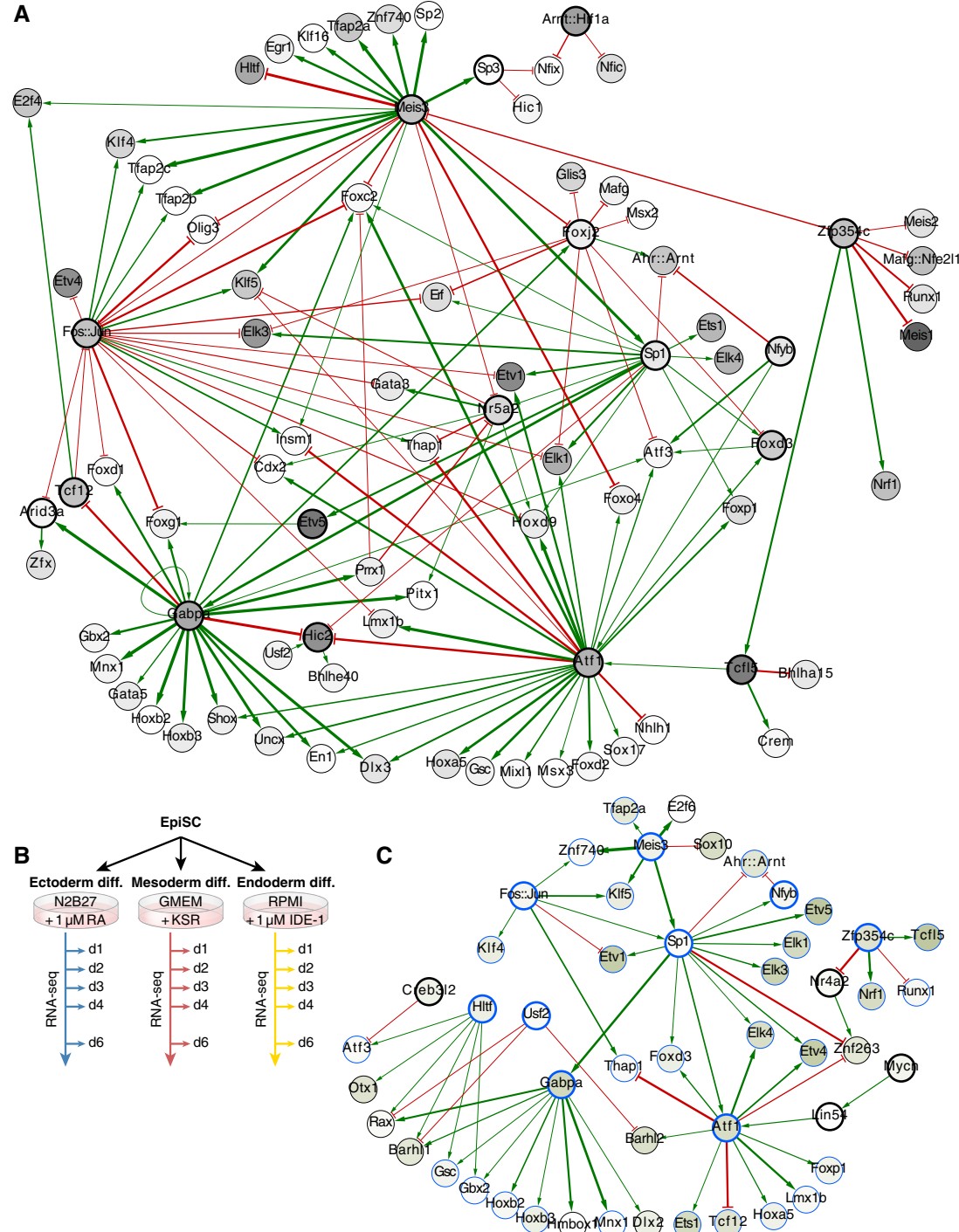

**Figure 4. Gene regulatory networks underlying mESC and EpiSC differentiation.**

A  Gene regulatory network underlying mESC differentiation to the three germ layers. Gray node shades indicate the significance of the motif activity, and edge thickness indicates the strength of the interaction.

B  Scheme of the experimental approach to quantitatively capture gene expression changes during EpiSC differentiation toward the three germ layers.

C  Gene regulatory network underlying EpiSC differentiation to the three germ layers. Green node shades indicate the significance of the motif activity, and edge thickness indicates the strength of the interaction. Nodes circled in blue are shared between the mESC and EpiSC differentiation networks.

network were target nodes. Among the highly connected nodes, Meis3, Sp1, and Fos:Jun stood out as being input nodes on other highly connected nodes. Altogether, these results demonstrated that

germ layer specification from a primed state is reminiscent of mESC differentiation, albeit being based on a more simplified transcriptional program.

The involvement of the same highly connected nodes in both mESC and EpiSC differentiation networks pointed to the potential important role of these nodes during fate specification. Indeed, their motif activities changed either in kinetics or in amplitude across the three differentiation procedures (Fig EV3E–L). This would effectively lead to the differential regulation of the target nodes under different signaling conditions.

**Simultaneous monitoring of germ layer acquisition**

We set out to test the hypothesis that interfering with highly connected nodes might disturb the balance between fate acquisition. To do so, we developed a triple knock-in (3KI) mESC line with fluorescent reporters for ectoderm, mesoderm and endoderm formation (Fig 5A). We used as a starting point the widely used *Sox1*-GFP knock-in mESC line that reports on ectoderm formation (Aubert *et al*, 2003). T (also known as Brachyury), an established marker of mesoderm and endoderm formation (Kubo *et al*, 2004), was targeted following the strategy reported in Fehling *et al* (2003). We selected as non-overlapping fluorescent reporter H2B-3xTagBFP containing the second intron of the mouse β-actin gene (Fig EV4A). Finally, eomesodermin (Eomes) served as a marker of definitive endoderm (Teo *et al*, 2011) by targeting H2B-mCherry to its locus (Fig EV4B).

Ectoderm differentiation led to GFP$^+$ cells that never expressed TagBFP that reports on Brachyury expression, a marker of mesendoderm and EpiSCs (Figs 5B and EV4C and D). Conversely, mesoderm differentiation led to a majority of TagBFP$^+$ cells (Figs 5C and EV4E and F), while Cherry$^+$ cells arose from TagBFP$^+$ cells (Figs 5D and EV4G). In addition, we determined differentiation conditions enabling the concomitant formation of GFP$^+$, TagBFP+, or Cherry$^+$ cells (Fig 5E, see Materials and Methods). In fact, the 3KI line showed that TagBFP and GFP expression (reflecting on Brachyury and Sox1 expression, respectively) were largely mutually exclusive (Fig 5F). This suggested the existence of a tipping point when differentiating mESC cells choose between endoderm or ectoderm specification. Moreover, this validated the fact that EpiSCs did not constitute a common differentiation intermediate. More importantly, the 3KI line enables us to probe in a quantitative manner the influence of the highly connected nodes on cell fate decision making.

**Probing the mESC differentiation network**

In order to quantitatively assess the role in differentiation of seven highly connected nodes, we conducted CRISPR/Cas9-mediated knockout of each of them in the 3KI mESC line. Three

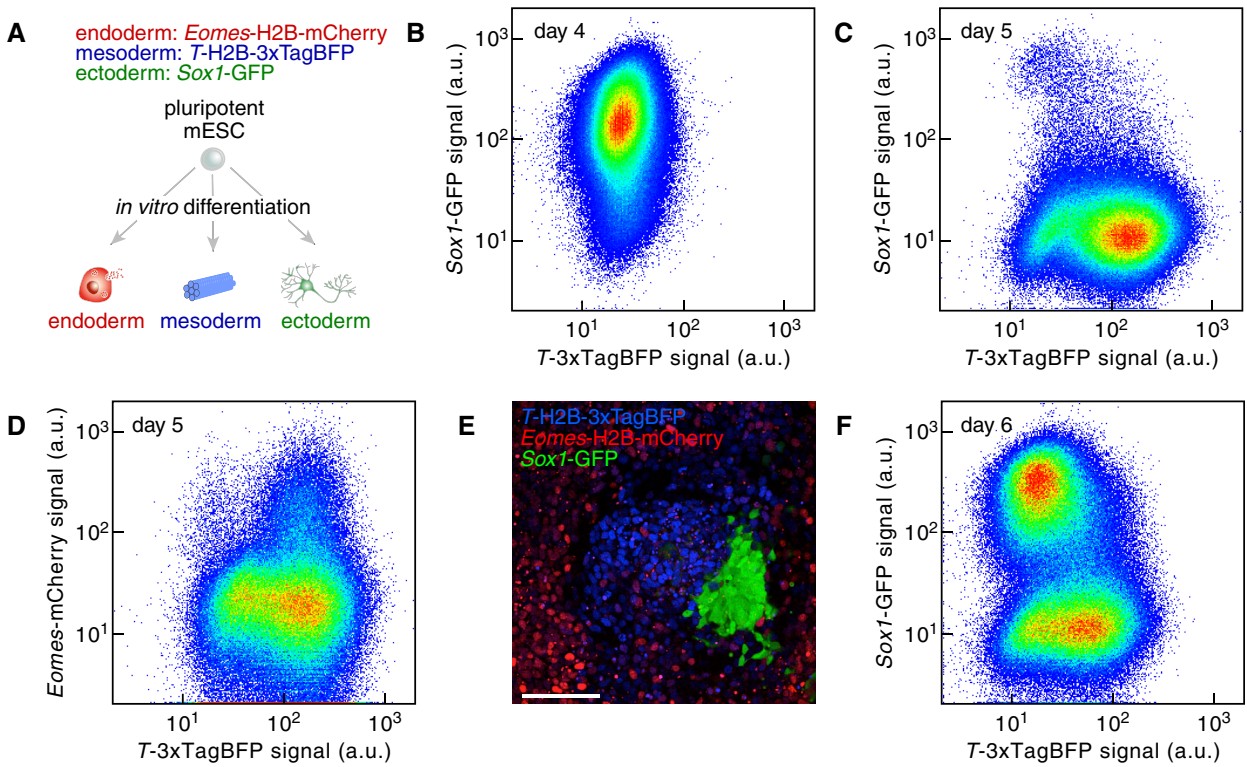

**Figure 5.  A *Sox1-Brachyury-Eomes* triple knock-in mESC line to simultaneously monitor germ layer specification.**

A   Triple knock-in (3KI) mESC line that simultaneously reports on ectoderm fate, (marked by the expression of *Sox1*::GFP), mesendoderm fate (marked by the expression of *T*::3xTagBFP, T is also known as *Brachyury*), or definitive endoderm fate (marked by the expression of *Eomes*::mCherry).

B   Ectoderm differentiation of the 3KI mESC line (marked by the expression of *Sox1*::GFP).

C   Mesendoderm differentiation of the 3KI mESC line (marked by the expression of *T*::3xTagBFP).

D   Endoderm differentiation of the 3KI mESC line (marked by the expression of *Eomes*::mCherry).

E, F   Spontaneous differentiation of the 3KI mESC line leading to the coexistence of cells belonging to the three germ layers. Scale bar: 100 μm.

scenarios are possible: A node can impact the acquisition of a specific fate, the balance between fates or differentiation in general (Fig 6A). We obtained $Atf1^{-/-}$, $Fos^{-/-}Jun^{-/-}$, $Foxj2^{-/-}$, $Meis3^{-/-}$, $Nr5a2^{-/-}$, $Sp1^{+/-}$, and $Zfp354c^{-/-}$ mESCs in the 3KI line background (Fig EV5). These lines were systematically differentiated to ectoderm, mesoderm, and endoderm, and the 3KI reporter signal was measured by flow cytometry to gauge fate acquisition. The loss of $Sp1$, $Meis3$, and $Zfp354$ favored the upregulation of GFP upon ectoderm differentiation while the knockout of $Nr5a2$ severely impaired it (Fig 6B). Upon mesendoderm differentiation, the fraction of TagBFP-positive cells increased in $Sp1^{+/-}$ and $Fos^{-/-}Jun^{-/-}$ cells, while $Zfp354c^{-/-}$ cells switched on Brachyury at a reduced frequency (Fig 6C). Finally, $Sp1$, $FosJun$, $Nr5a2$, and $FoxJ2$ depletion led to an increase in Cherry-positive cells compare to WT cells (Fig 6D). Taken together, the loss of $Sp1$ increased the fraction of cells positive for marker gene expression upon differentiation, implying that Sp1 was an inhibitor of mESC differentiation. Moreover, the formation of TUJ1-positive cells was severely impaired upon neuroectoderm differentiation of $Nr5a2^{-/-}$ mESCs (Appendix Fig S3).

While genes could have independent effects on each fate acquisition, another possibility is that differentiating mESCs choose one fate among several accessible fates. We therefore turned to conditions under which cells can spontaneously differentiate to ectoderm or mesendoderm fate as measured by the expression of Sox1 or Brachyury (Fig 6E). Under these conditions, ectoderm differentiation was favored for WT mESCs (Fig 6F). $Sp1^{+/-}$ mESCs differentiated

like WT mESCs, suggesting that $Sp1$ depletion, while increasing the numbers of marker-positive cells in individual fate acquisition, is neutral with respect to the choice between ectoderm and mesendoderm fates (Fig 6F). As expected, $Nr5a2^{-/-}$ mESCs that have impaired ectoderm differentiation capabilities did not upregulate GFP but instead switched on TagBFP in a fraction of the cells (Fig 6F). Surprisingly, $Fos^{-/-}Jun^{-/-}$ cells upregulated TagBFP in the vast majority of the population, with a small fraction of $Sox1^+$-positive cells being also Brachyury positive (Fig 6F). Thus, Fos:Jun activity positively biases mESCs to differentiate toward ectoderm.

# Discussion

We identified a common differentiation program comprising ~3,000 genes that drove exit from self-renewal and pluripotency. The exit from naïve pluripotency was accompanied by striking changes in cell–cell communication and cell–matrix architecture. While mESC self-renew as three-dimensional colonies, their differentiated derivatives form monolayers irrespective of their germ layer identity. These structural changes are orchestrated by the upregulation of extracellular matrix proteins and key EMT players. A second regulatory layer—acting in parallel to the loss of pluripotency and self-renewal—drove the divergence of trajectories into separate valleys. Surprisingly, this diversification of gene expression into lineage-specific programs occurred already 24 h after the application of differentiation cues.

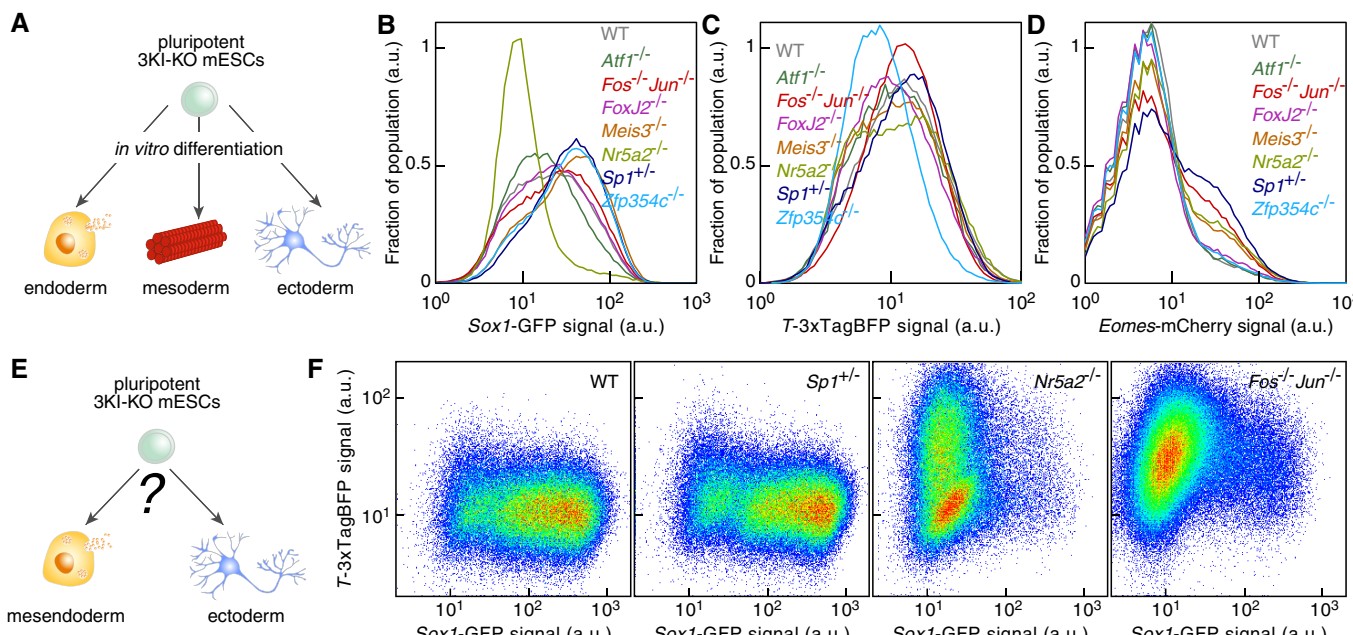

**Figure 6. Determining the role of the highly connected nodes on mESC differentiation.**

A   Scheme to assess the influence of the highly connected nodes on germ layer acquisition.
B   Differentiation of $Atf1^{-/-}$, $Fos^{-/-}Jun^{-/-}$, $Foxj2^{-/-}$, $Meis3^{-/-}$, $Nr5a2^{-/-}$, $Sp1^{+/-}$, and $Zfp354c^{-/-}$ mESCs in the 3KI background to ectoderm.
C   Differentiation of $Atf1^{-/-}$, $Fos^{-/-}Jun^{-/-}$, $Foxj2^{-/-}$, $Meis3^{-/-}$, $Nr5a2^{-/-}$, $Sp1^{+/-}$, and $Zfp354c^{-/-}$ mESCs in the 3KI background to mesendoderm.
D   Differentiation of $Atf1^{-/-}$, $Fos^{-/-}Jun^{-/-}$, $Foxj2^{-/-}$, $Meis3^{-/-}$, $Nr5a2^{-/-}$, $Sp1^{+/-}$, and $Zff354c^{-/-}$ mESCs in the 3KI background to definitive endoderm.
E   Scheme to assess the influence of the highly connected nodes on the spontaneous fate acquisition between mesendoderm and ectoderm.
F   Differentiation of wild type (WT), $Sp1^{+/-}$, $Nr5a2^{-/-}$, and $Fos^{-/-}Jun^{-/-}$ mESCs in conditions under which cells can acquire mesendoderm or ectoderm fates.

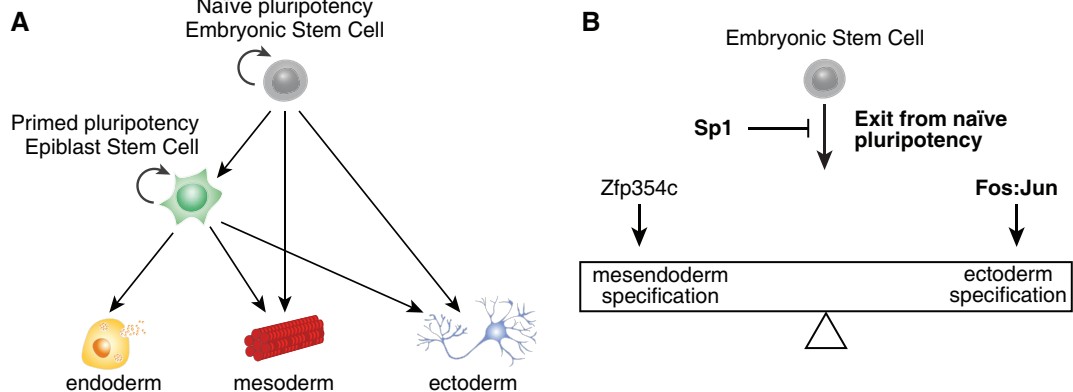

**Figure 7. Model of germ layer specification from mESCs.**

A Model showing the hierarchy between naïve and primed pluripotency as well as germ layer specification.

B Regulators of the balance between mesendoderm and ectoderm specification at the exit of pluripotency.

While the primed pluripotency of the postimplantation epiblast is considered as the common origin of germ layer specification *in vivo* (Murry & Keller, 2008; Nichols & Smith, 2009), EpiSCs do not constitute an obligatory intermediate *in vitro*. EpiSCs stably captured an intermediate state of endodermal differentiation that was not shared with ectodermal or mesodermal differentiation trajectories (Fig 7A). It should be noted that EpiSCs are pluripotent and that their differentiation potential is therefore not restricted by the expression of germ layer markers. The similarities between EpiSC differentiation and initial stages of endoderm are likely due to their shared dependency on Activin signaling (Brons *et al*, 2007; Tesar *et al*, 2007; Borowiak *et al*, 2009).

Spatially defined transcriptomes of posterior regions of the epiblast that constitute the primitive streak were similar to the expression profiles of endoderm differentiation intermediates and EpiSCs. In fact, EpiSCs and the primitive streak share functional properties (Kojima *et al*, 2014). The other regions of the mouse epiblast have expression profiles that are distinct from the ones of primed pluripotency and the primitive streak and thus might be closer to the state of formative pluripotency (Smith, 2017). This is consistent with the regionalization of the epiblast into different progenitors of the germ layers (Tam & Behringer, 1997). Notably, ectoderm is specified from the anterior epiblast (Tam & Behringer, 1997), which never transits through a primitive streak-like state (Peng *et al*, 2019). Overall, *in vitro* germ layer differentiation mirrored remarkably lineage specification *in vivo*.

The gene regulatory network governing mESC differentiation was devoid of feedback loops, in contrast with the regulatory networks supporting pluripotency (Chen *et al*, 2008; Kim *et al*, 2008). Feedback loops are indeed used to stabilize particular cellular states (Alon, 2007). Instead, the mESC differentiation network relied on dense overlapping regulon motifs (Alon, 2007), hinting that mESCs integrate several inputs during fate acquisition. The network regulating EpiSC differentiation was a simplified version of the mESC network, in accordance with the more differentiated character of primed pluripotency compared to naïve pluripotency.

The triple knock-in *Sox1-Brachyury-Eomes* mESC line we developed allows the direct readout of the proportion of cells belonging to specific germ layer fates. Therefore, it facilitates the quantitative exploration of fate acquisition after genetic perturbations. The highly connected nodes in the mESC differentiation network were not previously reported to play a role in mESC differentiation. Using CRISPR/Cas9 knockouts in the 3KI mESC line, we showed that these nodes had diverse influences on the acquisition of differentiated fates (Fig 7B). Sp1 inhibited differentiation and was neutral regarding fate choice. Deletion of Nr5a2 severely impaired ectoderm differentiation. Finally, Fos:Jun favored the acquisition of ectodermal fate at the expense of endoderm, while Zfp354c had the opposite effect. Some of these genes were shown to have important roles in other developmental contexts. For example, $Sp1^{-/-}$ mouse embryos are retarded in development (Marin *et al*, 1997). While we revised this manuscript, Sp1 was identified as an important factor for lineage specification during mouse gastrulation and *Sp1* knockout in mESCs facilitated the exit of naïve pluripotency (Peng *et al*, 2019), corroborating our findings. Nr5a2 plays a critical role at later stages of neural development (Stergiopoulos & Politis, 2016). Ectopic Jun expression resembles retinoic acid treatment of embryonal carcinoma cells (de Groot *et al*, 1990). This parallels our findings that Fos:Jun positively biases the acquisition of ectoderm fate, a differentiation that relies on treatment with retinoic acid.

In conclusion, we identified novel regulators of mESC differentiation by inferring a gene regulatory network form deep sequencing data at high temporal resolution. The role of these genes was established using CRISPR/Cas knockouts in a multicolor fluorescent reporter mESC line. Importantly, the fact that naïve cells are competent to engage directly in lineage decision making without passing through a primed state stresses the need for a comparative study of the acquisition of several fates in order to determine the rules of mESC differentiation.

## Materials and Methods

### mESC maintenance and differentiation

mESCs were R1 (Nagy *et al*, 1993) (a kind gift by the EMBL Heidelberg Transgenic Services) or E14TG2a (ATCC CRL-1821). mESCs

were maintained in "2i" or "LIF+serum" as described previously (Sladitschek & Neveu, 2015b).

mESCs were differentiated toward an endodermal progenitor fate following the protocol described by Borowiak et al (2009). Briefly, mESCs were seeded at a density of 2,500 cells per cm$^2$ onto 0.1% gelatin-coated dishes 1 day prior to the start of the differentiation procedure. The following day, cells were rinsed in D-PBS and switched to endodermal differentiation medium (Advanced RPMI 1640 (Thermo Fisher), 1 µM IDE-1 (Tocris), 0.2% (v/v) fetal calf serum (Millipore), 2 mM L-glutamine (Sigma)). Samples collected 24 h after switching to the differentiation regime are referred to as "day 1" differentiation samples. Medium was replaced every day.

mESCs were differentiated toward a mesodermal progenitor fate following the protocol described in Torres et al (2012). Briefly, mESCs were seeded at a density of 2,500 cells per cm$^2$ onto 0.1% gelatin-coated dishes 1 day prior to the start of the differentiation procedure. The following day, cells were rinsed in D-PBS and switched to mesodermal differentiation medium [Glasgow's MEM (Thermo Fisher), 10% (v/v) KnockOut Serum Replacement (Thermo Fisher), 0.1 mM 2-mercaptoethanol (Invitrogen), 1× non-essential amino acids (Gibco), and 1 mM sodium pyruvate (Gibco)]. Medium was replaced every day.

mESCs were differentiated toward a neuroectodermal progenitor fate following the protocol developed by Ying et al (2003). Briefly, mESCs were seeded at a density of 7,500 cells per cm$^2$ onto 0.1% gelatin-coated dishes 1 day prior to the start of the differentiation procedure. The following day, cells were washed in D-PBS and switched to N2B27 medium (N2B27 medium was prepared from a 1:1 mixture of DMEM/F12 (without HEPES, with L-glutamine) and neurobasal medium with 0.5× B-27 (with vitamin A) and 0.5× N-2 supplements, 0.25 mM L-glutamine, 0.1 mM 2-mercaptoethanol (all Invitrogen), 10 µg/ml BSA fraction V, and 10 µg/ml human recombinant insulin (both Sigma)). all-trans-Retinoic acid (Sigma) was added at 1 µM to the differentiation medium 24 h after the start of the differentiation procedure. Medium was replaced every other day.

mESCs were differentiated toward EpiSCs as described (Guo et al, 2009). Briefly, mESCs were seeded at a density of 50,000 cells per cm$^2$ onto 0.1% gelatin-coated dishes 1 day prior to the start of the differentiation procedure. The following day, cells were washed in D-PBS and switched to N2B27 medium (prepared using B27 supplement without vitamin A) supplemented with 12 ng/µl FGF2 and 20 ng/µl Activin (both Peprotech). Medium was replaced every day, and cells were passaged every other day using 0.05% Trypsin (Invitrogen).

For spontaneous differentiation between mesendoderm and ectoderm fates, mESCs were seeded at a density of 10,000–20,000 cells per cm$^2$ onto 0.1% gelatin-coated dishes 1 day prior to the start of the differentiation procedure. The following day, cells were rinsed in D-PBS and switched to endodermal differentiation medium (Advanced RPMI 1640 (Thermo Fisher), 1 µM IDE-1 (Tocris), 0.2% (v/v) fetal calf serum (Millipore), 2 mM L-glutamine (Sigma)). Samples collected 24 h after switching to the differentiation regime are referred to as "day 1" differentiation samples. Medium was replaced every day.

## Immunostaining

Cells were grown on µ-slides (Ibidi) and fixed in 4% paraformaldehyde (PFA) in D-PBS (without Ca$^{2+}$ and Mg$^{2+}$) for 10 min at RT followed by PFA inactivation in 300 mM glycine in D-PBS (5 min, RT)

and a wash in D-PBS. Cells were permeabilized with 1% (v/v) Triton X-100, 0.2% (w/v) SDS, 10 mg/ml BSA in D-PBS (1 h, RT) and incubated with primary antibodies overnight at 4°C in 50 mg/ml BSA in TNT (100 mM Tris–Cl (pH 7.5), 150 mM NaCl, and 0.1% (v/v) Tween-20). The following antibodies and dilutions were used: rabbit anti-TUJ1 (Cell Signaling, 5568) at 1:600; rabbit anti-DESMIN (Cell Signaling, 5332) at 1:300; rabbit anti-GATA6 (Cell Signaling, 5851) at 1:1,600. An anti-rabbit IgG (H+L) F(ab')$_2$ Fragment Alexa Fluor 647 Conjugate (Cell Signaling, 4414) served as secondary antibody and was allowed to incubate for 2 h at RT in 50 mg/ml BSA in TNT. Nuclei were visualized using a constitutive nuclear marker (a stably integrated CAG::H2B-mCherry-BGHpA plasmid). Confocal images were acquired on an inverted SP8 confocal microscope (Leica) equipped with a 40× PL Apo 1.1 W objective. TUJ1 immunostaining for quantification by flow cytometry was performed under the same conditions except that the starting material was a single-cell suspension.

## RNA-Seq library construction

RNA was extracted from pellets of trypsinized cells using the MirVana kit (Ambion) following the instructions provided by the manufacturer. For each time point of the differentiation procedure, two independent biological replicates were analyzed. Total RNA samples of mouse organs (lung, liver, brain, heart, kidney, smooth muscle, spleen, thymus) (referred to herein as "differentiated tissues") and of E7 mouse embryos were purchased from Clontech. Sixty-nine barcoded mRNA libraries were prepared using TruSeq RNA Sample Preparation (Illumina) following the manufacturer's instructions. Libraries were run on Illumina HiSeq 2000 in the 50SE regime. Sequencing results are available on ArrayExpress with accession E-MTAB-4904. In addition, we used mRNA expression data that we previously deposited on ArrayExpress with accession E-MTAB-2830 and E-MTAB-3234.

## RNA-Seq analysis

Ensembl cDNAs of the mouse genome release GRCm38 were masked with RepeatMasker (Smit, AFA, Hubley, R and Green, P. Repeat-Masker Open-3.0. 1996–2010 http://www.repeatmasker.org), and a Bowtie index was built using these masked transcripts. Reads were aligned to this index using Bowtie (Langmead et al, 2009) with default parameters. mRNA read counts were determined for each Ensembl ID using custom Python scripts. Read counts were not normalized by the transcript length for individual genes as we were solely interested in relative expression changes across samples. Read counts were first grossly normalized to account for different sequencing depth by correcting for the total number of aligned reads. A finer normalization factor was then determined by matching median-filtered log-transformed read counts to the identity line for genes that are highly expressed in all samples. For clustering analysis, we kept genes with a maximal expression > 3 reads per million across samples and at least a fourfold variation in expression. Principal component analysis was carried out as described in Neveu et al (2010).

## Inference of the mESC and EpiSC gene regulatory networks

The upstream 1 kb proximal region from the start site of mouse protein coding genes was retrieved from Ensembl. We predicted binding sites for transcription factors using their weight matrices

from the curated JASPAR database (Mathelier *et al*, 2016). We kept transcription factors with a maximal expression > 6 reads per million across samples maintained in "LIF+serum" and differentiated to endoderm, mesoderm and ectoderm. The computation of their motif activity was done following the framework described in Bussemaker *et al* (2001). Transcription factors which motifs had a statistical significance < 0.01 [as defined in Bussemaker *et al* (2001)] were considered as nodes of the network. We added an edge between two nodes if the two following conditions were fulfilled: (i) a predicted binding site of the regulator in the 1 kb promoter region of the target node, (ii) the absolute value of the correlation coefficient between the regulator motif activity and the target gene expression profile was greater than 0.8. The interaction was considered inductive if the correlation coefficient was positive and inhibitory if the correlation coefficient was negative. The network was visualized using Cytoscape. The derivation of the EpiSC differentiation network was done similarly using samples from day 9 onwards of the EpiSC differentiation when transcription profiles have stabilized and EpiSCs differentiated to endoderm, mesoderm, and ectoderm.

### Analysis of embryo-derived EpiSC gene expression profiles from Tesar *et al* (2007)

To compare gene expression profiles of *in vitro* derived EpiSCs with the ones of embryo-derived EpiSCs, we used data from Tesar *et al* (2007) with accession number GSE7866. The distribution of microarray probe signals was quantile normalized to adjust it to the distribution of read counts from our mRNA-Seq data. Principal components were computed using only the expression profiles of our *in vitro* endoderm, mesoderm, and ectoderm differentiation and keeping genes (represented with probes in the microarray data) with a 16-fold variation in expression during endoderm, mesoderm, or ectoderm differentiation. Gene expression profiles from our data (mESCs and EpiSC differentiation from day 8 onwards) and from Tesar *et al* (2007) (mESCs and embryo-derived EpiSCs) were projected on the first two principal components.

### Analysis of 3D transcriptomes of mouse gastrulation from Peng *et al* (2019)

To compare our *in vitro* differentiation data with the gastrulating mouse embryo, we used the Geo-Seq data from Peng *et al* (2019) with accession number GSE120963. Gene expression levels were log2-transformed. For each gene, the mean expression level across samples was subtracted to correct for batch effect between mRNA-Seq and Geo-Seq. Principal components were computed using only the expression profiles of our *in vitro* endoderm, mesoderm, and ectoderm differentiation and keeping genes with a 16-fold variation in expression during endoderm, mesoderm, or ectoderm differentiation and an eightfold variation in expression between different sections of E7.5 embryos. Geo-Seq expression profiles of E5.5, E6.0, E6.5, E7.0, and E7.5 embryos were projected on the first two principal components.

### Generation of a *Sox1-Brachyury-Eomes* triple knock-in mESC reporter line

We used as starting point an established *Sox1*-GFP reporter line (Aubert *et al*, 2003) (a kind gift of Austin Smith). T (also known as

Brachyury), an established marker of mesoderm and endoderm formation (Kubo *et al*, 2004), was targeted using the strategy reported in Fehling *et al* (2003) with a non-overlapping fluorescent reporter H2B-3xTagBFP containing the second intron of the mouse β-actin gene (Sladitschek & Neveu, 2015b). Eomesodermin (Eomes), a marker of definitive endoderm (Teo *et al*, 2011), was targeted with an H2B-mCherry reporter. We first assembled a Neo/Kan resistance cassette flanked by FRT sites that is compatible with the MXS-chaining strategy (Sladitschek & Neveu, 2015a). Targeting constructs were generated by standard ET recombineering (Muyrers *et al*, 1999; Zhang *et al*, 2000; Wang *et al*, 2006) using BACs for Brachyury (bMQ-343F18) and Eomes (bMQ-421D6) (Adams *et al*, 2005). Targeting constructs were linearized and transfected using Fugene HD (Promega) according to the manufacturer's protocol. After antibiotic selection, single colonies were expanded and screened for correct targeting. Targeting was confirmed by genomic PCR or Southern blotting carried out using $^{32}$P-labeled RNA probes following Church and Gilbert (1984). After each round of targeting, the line was subcloned after removal of the selection cassette by transient transfection of pPGKFLPobpA (Addgene plasmid 13793, a kind gift of Philippe Soriano) (Raymond & Soriano, 2007).

### Generation of *Atf1*$^{-/-}$, *Fos*$^{-/-}$*Jun*$^{-/-}$, *Foxj2*$^{-/-}$, *Meis3*$^{-/-}$, *Nr5a2*$^{-/-}$, *Sp1*$^{+/-}$, and *Zfp354c*$^{-/-}$ mESCs

RNA-guided Cas9 nucleases were used to introduce inactivating mutations in the following genes: *Atf1*, *Fos* and *Jun*, *Foxj2*, *Meis3*, *Nr5a2*, *Sp1* or *Zfp354c*. Guide RNA inserts targeting the third exon of *Atf1* resulting in a non-functional protein (Bleckmann *et al*, 2002) (with genome target sequence: 5′-GCTGCTCGTCTGATAGATGG), the second exon of *Fos* deleting the leucine zipper (5′-GACTGGGTGGGGAGTCCGTA), the beginning of *Jun* exon deleting the leucine zipper (5-GGTCCGAGTTCTTGGCGCGG), the first exon of *Foxj2* (5′-GAGCACTTCCGGGCGCCCCC), the first exon of *Meis3* (5′-GATGAGCTGCGCCACTACCC), the fourth exon of *Nr5a2* deleting the DNA-binding domain resulting in a non-functional protein (Botrugno *et al*, 2004; Gu *et al*, 2005) (5′-GTGTGTGGCGATAAAGTGTC), the third exon of *Sp1* deleting the DNA-binding domain (Marin *et al*, 1997) (5′-GACCATTAACCTCAGTGCAT), and the third exon of Zfp354c deleting the zinc finger domains (5′-GTGATTGGCAAGCTGCAAAA) were designed and cloned in pX330-U6-Chimeric-BB-CBh-hSpCas9 following Hsu *et al* (2013). The Cas9 plasmids were transfected in the 3KI mESC line. Successfully edited clones corresponding to *Atf1*$^{-/-}$, *Fos*$^{-/-}$*Jun*$^{-/-}$, *Foxj2*$^{-/-}$, *Meis3*$^{-/-}$, *Nr5a2*$^{-/-}$, *Sp1*$^{+/-}$, and *Zfp354c*$^{-/-}$ mESCs were validated by genomic PCR.

### Flow cytometry and fluorescence-activated cell sorting (FACS)

Cells were trypsinized and dissociated to single-cell suspension. Cells were pelleted at 1,000 *g* for 1 min, resuspended in D-PBS, and strained through a 40-μm filter. Cells were analyzed on an LSRFortessa flow cytometer (BD BioSciences). GFP-positive and GFP-negative cells as well as TagBFP-positive and TagBFP-negative cells were FACS-purified using a MoFlo sorter (DakoCytomation) or an Aria Fusion sorter (BD BioSciences) during the differentiation of the *Sox1-Brachyury-Eomes* triple knock-in mESC reporter line. Flow cytometry data were analyzed with FlowJo or custom Python scripts.

## Western Blot

The following primary antibodies were used at the indicated dilutions under agitation overnight at 4°C: mouse anti-ATF1 (25C10G) (Santa Cruz Biotechnology, sc-270) at 1:100, rabbit monoclonal anti-c-FOS (9F6)(Cell Signaling, 2250) at 1:1,000, mouse monoclonal anti-FOXJ2 (G-9) (Santa Cruz Biotechnology, sc-514265) at 1:200, mouse monoclonal anti-GAPDH (D4C6R) (Cell Signaling, 97166) at 1:10,000, rabbit monoclonal anti-GAPDH (D16H11) (Cell Signaling, 5174) at 1:400,000, rabbit monoclonal anti-c-JUN (60A8) (Cell Signaling, 9165) at 1:1,000, rabbit polyclonal anti-MEIS3 (Proteintech, 12775-1-AP) at 1:2,000, rabbit polyclonal anti-NR5A2 (Abcam, ab189876) at 1:2,000, mouse monoclonal anti-SP1 (E-3) (Santa Cruz Biotechnology, sc-17824) at 1:100, rabbit polyclonal anti-ZNF354C (biorbyt, orb1661) at 1:2,000. The following secondary antibodies were used at the indicated dilutions under agitation for 2 h at RT: goat anti-mouse IgG (H+L) conjugated to horseradish peroxidase (Jackson ImmunoResearch, 115-035-146) at 1:50,000–1:200,000, goat anti-rabbit IgG (H+L) conjugated to horseradish peroxidase (Jackson ImmunoResearch, 111-035-144) at 1:80,000–1:200,000. Blocking and incubation with antibodies were carried out in 50 mg/ml BSA in TNT-buffer (100 mM Tris–Cl pH 7.5, 150 mM NaCl, 0.1% (v/v) Tween-20). Horseradish peroxidase was poisoned by incubating in 10 mM sodium azide in TNT for 1 h at RT.

## Statistical analysis

Statistical tests were computed using the Python SciPy module. When appropriate, we corrected for multiple hypothesis testing following Benjamini and Hochberg (1995). The *P*-value associated with the enrichment of a specific gene set among a larger gene pool was estimated from an enrichment distribution determined from > 10,000 re-samplings of the larger gene pool.

# Data availability

Sequencing results are available on ArrayExpress (https://www.ebi.ac.uk/arrayexpress/) with accession E-MTAB-4904.

**Expanded View** for this article is available online.

## Acknowledgements
This work was technically supported by the EMBL Flow Cytometry Core and Genomics Core facilities. The study was funded by EMBL.

## Author contributions
HLS and PAN conceived the study, carried out experiments, analyzed the data, and wrote the paper.

## Conflict of interest
The authors declare that they have no conflict of interest.

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
