## [Review Process File · Molecular Systems Biology]

A gene regulatory network controls the balance between mesendoderm and ectoderm at pluripotency exit

Hanna L Sladitschek, Pierre A Neveu.

Review timeline:

Submission date:	6 th June 2019
Editorial Decision:	26 th July 2019
Revision received:	1 st August 2019
Editorial Decision:	2 nd August 2019
Revision received:	8 th October 2019
Editorial Decision:	10 th November 2019
Revision received:	11 th November 2019
Accepted:	14 th November 2019

Editor: Jingyi Hou

Transaction Report:

1st Editorial Decision

26th July 2019

Thank you for submitting your work to Molecular Systems Biology. We have now heard back from the two referees who agreed to evaluate your manuscript. As you will see below, while the referees acknowledge that the topic of the study is relevant, they raise significant concerns about your work, and both referees indicated that they do not support publication in Molecular Systems Biology. Under these circumstances and considering the overall rather low level of support provided by the referees, I see no other choice than to return the manuscript with the message that we cannot offer to publish it.

I am sorry not to be able to bring better news on this occasion. In any case, thank you for the opportunity to examine your work and I hope that the points raised in the reports will prove useful to you.

REFeree REPORTS

Reviewer #2:

The differentiation of mouse ES cells in adherent culture has been the subject of intense scrutiny for the last ten years so, any piece of work attempting to make a significant contribution has a very high bar to rise to. In this report, Sladitschek and Neveau claim to make such a contribution. Unfortunately the report, while containing a fair amount of work and information, it falls very short of the mark. The main problem is that the culture conditions are ill defined and do not reflect the normal differentiation of the cells in vivo but even if the conditions were right, there is an ample literature on the subject that the authors ignore (some examples below) and which should have been discussed to provide evidence for novelty and insight. Furthermore, it is surprising that with the recent flurry of single cell papers on the early postimplantation development of the mouse, the authors do not make any attempt to check their results against them. There is also much that is

misleading, in particular their interpretation of EpiSCs and the experiments they do with them. Overall, while I appreciate the work and care that has gone into this work, there is a naivete about the field and an ignorance of the literature that cannot be ignored.

In terms of the protocols, the authors use old and ill defined conditions; there are plenty of reports which they should look at that use better conditions closer to the embryo one see e.g Loh et al. (2014) Efficient endoderm induction from human pluripotent stem cells by logically directing signals controlling lineage bifurcations PMID 24412311 and if they go to PubMed, they can find many related references. What is also concerning is statements like the expression of Gata 6 six days into differentiation is a reflection of endoderm differentiation when, actually, at that time (and the pathways of in vitro adherent differentiation are very well outlined) is a marker of cardiac differentiation. If they have any trace of endoderm, they should check in a more rigorous manner. Their apparently surprising result that EpiSCs can only be differentiated into endoderm (under their experimental conditions) is easily explained by the realization that they culture the cells in Activin and FGF which will direct the cells into endoderm. EpiSCs can be differentiated into the three germ layers (the literature is ample but here two examples: Kojima et al. (2014) the transcriptional and functional properties of mouse epiblast stem cells resemble anterior primitive streak PMID 24139757 and Edri et al. (2019) An epiblast stem population derived multipotent population for axial extension PMID 31023877

The network analysis is fine, though neither novel nor insightful but it is well done; unfortunately given the problems with the data, the output reflects the response of the cells to the culture conditions rather than what actual happens in vivo. The genetic loss of function experiments follow suit.

The basic idea is good and the reporter and strategy are all fine but the culture conditions are very artificial and we all know that cells will respond to cocktails of signals in interesting manners.

Reviewer #3:

The manuscript of Sladitschek and Neveu describes the identification of regulators of mouse ES cells differentiation.

In particular they showed that the EpiSC state does not represent a common intermediate between ES cells and germ layers specification in vitro but an intermediate state of endodermal differentiation. Moreover, by using motif activity computational analysis they defined a transcriptional network governing mouse ES cells differentiation and the essential nodes which were validated by using CRISPR/Cas9 knockouts in a multicolor fluorescence reporter system. Finally they identified Sp1 as a general inhibitor of differentiation, Nr5a2 as a regulator of ectoderm differentiation and Fos:Jun and Zfp354c as switches between ectoderm and mesoderm fate.

Overall the results are potentially interesting but the conclusion that EpiSC represent a locked state along the trajectory of endoderm al differentiation is at odds with previously published results of in-vivo derived EpiSCs and the general knowledge in the field (Kojima et., 2014 Cell Stem Cell). Moreover, the generation of a fluorescent reporter line is potentially useful to monitor germ layers specification during differentiation of mouse ES cells although some experimental validations need to be performed.

In order to improve the impact of this work I advise the authors to address the following points:

1) the authors concluded that EpiSCs represent an intermediate state of endodermal differentiation. To demonstrate the correct differentiation of mES cells into EpiSCs in vitro they should show expression of markers of naïve and primed pluripotency in their in-vitro derived EpiSCs (Figure 3A). Also, In figure 3B and 3C they should include published embryo-derived EpiSCs as control in the PCA and heatmap.

If EpiSCs remained locked at an intermediate state of differentiation, why are these cells used to differentiate towards the three germ layers in figure 4B and used to generate the gene regulatory network?

2) the authors generated a triple knock-in reporter line to monitor germ layer specification. However, they did not show specific up regulation of the three germ layer markers upon differentiation. First, the authors should show the basal activity of the 3 reporters in mouse ES cells under LIF+Serum conditions. It is currently hard to tell if the activity reported in figure 5 is basal/background or specific. Thus, in figure 5B-C-D fluorescence profiles of pluripotent mES should also be included.

Second, to confirm the specificity of their reporters, the authors should sort positive and negative reporter cells and show specific upregulation of mRNA of the relative marker (eg. GFP positive cells during ectoderm differentiation should show high levels of Sox1 as compared to GFP negative cells and undifferentiated cells).

Finally, spontaneous differentiation is used in figure 5E, 5F and 6E-F. How is this spontaneous differentiation achieved? Which culture conditions?

3) the KO generated by CRISPR/Cas9 have been validated by genomic PCR followed by sequencing, but the results presented are not clear. In 5 out of 8 lines the authors reported the presence of the same mutation on both alleles (indicated as "edited alleles"). This is very unlikely, as the most common outcome is the generation of two different mutant alleles. The authors should explain / present their results more clearly and an independent validation of their KO lines (by Western Blot / Immunostaining) would be preferable.

4) The authors propose, based on their reporter assays, that Nr5a2, Fos:Jun and Zfp354c are critical regulators of germ layer formation. Such claims are based only on reporter assay where 3 genes are monitored at a single point in time. For at least one of the genes identified it would be important to show that differentiation is actually impaired. For example, do Nr5a2 KO cells form TUJ1+ cells when exposed to Neuroectoderm differentiation protocol?

5) Minor points: In order to better dissect gene expression changes of mESC differentiation towards the three germ layers the authors should generate 2 venn diagrams plotting separately up and down regulated genes during differentiation of the 3 germ layers (figure 1C). The same analysis should be performed for heatmaps in figure 1D-E-F.

The hierarchical clustering in Figure2A show that already from day1 ectodermal cells clustered with differentiated tissues. Such a divergence is not clear in PCA analysis in Figure 2C.

Typos / mistakes are listed below:

Page1, first paragraph:

figure 1B is Figure 1A.

Figure 1C is Figure 1B

Page2, second paragraph:

Figure2E is mentioned but it is not present in the figures.

Page7, second paragraph, second line:

Is endoderm or mesoderm differentiation described?

Page7, second paragraph, fourth line:

"differentiation under endoderm promoting conditions" but in the figure legend is "spontaneous differentiation"

Page12, third paragraph:

"maximal expression >6" should be changed into "minimal expression >6"

We would like to appeal to your decision to reject our paper. Indeed, the major criticism of both reviewers that "EpiSCs are a locked state" relies on a misrepresentation of our findings in the case of reviewer 2 and on a clear misunderstanding of reviewer 3. In addition, we can address all the concerns raised by reviewer 3 as outlined below. Thank you for considering this request.

Reviewer #2:

The differentiation of mouse ES cells in adherent culture has been the subject of intense scrutiny for the last ten years so, any piece of work attempting to make a significant contribution has a very high bar to rise to. In this report, Sladitschek and Neveau claim to make such a contribution. Unfortunately the report, while containing a fair amount of work and information, it falls very short of the mark. The main problem is that the culture conditions are ill defined and do not reflect the normal differentiation of the cells *in vivo* but even if the conditions were right, there is an ample literature on the subject that the authors ignore (some examples below) and which should have been discussed to provide evidence for novelty and insight. Furthermore, it is surprising that with the recent flurry of single cell papers on the early postimplantation development of the mouse, the authors do not make any attempt to check their results against them.

Authors: We provide a detailed response to the comments that "the culture conditions are ill defined" and "there is an ample literature on the subject that the authors ignore" when the reviewer raises these points again below.

We will compare our findings to single-cell sequencing studies of early mouse postimplantation development. We can already state that the anterior primitive streak constitutes only a small fraction of the cells in the embryo (see for example Mohammed et al., Cell Reports 2017). The rest of the embryonic cells therefore follows differentiation trajectories that do not pass through a primitive streak state. This observation supports one of our key findings that EpiSCs (which resemble primitive streak cells) do not constitute an obligatory differentiation step.

There is also much that is misleading, in particular their interpretation of EpiSCs and the experiments they do with them. Overall, while I appreciate the work and care that has gone into this work, there is a naive about the field and an ignorance of the literature that cannot be ignored.

Authors: We find this comment particularly unfair as it relies both on a gross misrepresentation of our findings and on the reviewer overlooking data presented in the manuscript (as detailed below).

In terms of the protocols, the authors use old and ill defined conditions; there are plenty of reports which they should look at that use better conditions closer to the embryo one see e.g Loh et al. (2014) Efficient endoderm induction from human pluripotent stem cells by logically directing signals controlling lineage bifurcations PMID 24412311 and if they go to PubMed, they can find many related references.

Authors: Our aim is to study how mESCs respond to differentiation cues (using well established and widely used protocols that yield specific and functional cell types) and the relationship between the acquisition of different fates. The criticism that the conditions are "ill-defined" is difficult to understand. Notably, the ectoderm and mesoderm differentiations use chemically defined media and the endoderm differentiation is chemically induced.

What is also concerning is statements like the expression of Gata 6 six days into differentiation is a reflection of endoderm differentiation when, actually, at that time (and the pathways of *in vitro* adherent differentiation are very well outlined) is a marker of cardiac differentiation.

Authors: In fact, Gata6 is an established endoderm marker that is reported to be upregulated (along with other endoderm markers) by Borowiak et al. (2009) under the endoderm differentiation

conditions they pioneered, the very same conditions we use in this manuscript. In addition, Borowiak et al. (2009) report that "Endoderm induction from mouse ESCs peaks at day 6 of treatment with small molecules". In the absence of any specific literature reference, the assertion of the reviewer must therefore apply to another experimental system.

If they have any trace of endoderm, they should check in a more rigorous manner.

Authors: The reviewer completely ignores the fact that we carried out time-course RNA-Seq for the three differentiation protocols (among which endoderm differentiation), the immunostainings displayed in Figure 1A being for illustration purposes. We show a time course of expression of some endoderm markers in Supplementary Figure S3 and are happy to display data for more markers.

Their apparently surprising result that EpiSCs can only be differentiated into endoderm (under their experimental conditions) is easily explained by the realization that they culture the cells in Activin and FGF which will direct the cells into endoderm. EpiSCs can be differentiated into the three germ layers (the literature is ample but here two examples: Kojima et al. (2014) the transcriptional and functional properties of mouse epiblast stem cells resemble anterior primitive streak PMID 24139757 and Edri et al. (2019) An epiblast stem population derived multipotent population for axial extension PMID 31023877

Authors: The reviewer's statement "Their apparently surprising result that EpiSCs can only be differentiated into endoderm" is factually wrong and misleading and therefore lacks merit. In fact, we differentiated EpiSCs to the three germ layers (Figure 4B and C). In addition, we wrote page 6 (beginning of second paragraph): "While we found that in vitro differentiation protocols allow to bypass primed pluripotency, EpiSCs are themselves pluripotent".

The network analysis is fine, though neither novel nor insightful but it is well done; unfortunately given the problems with the data, the output reflects the response of the cells to the culture conditions rather than what actual happens in vivo. The genetic loss of function experiments follow suit.

The basic idea is good and the reporter and strategy are all fine but the culture conditions are very artificial and we all know that cells will respond to cocktails of signals in interesting manners.

Authors: The aim of our study is precisely to understand how cells respond to the different differentiation conditions (which are published and were shown to create functional cell types). Understanding mESC differentiation is of fundamental importance for regenerative medicine applications but insights gained from such studies might also be applicable to embryonic development.

Reviewer #3:

The manuscript of Sladitschek and Neveu describes the identification of regulators of mouse ES cells differentiation.

In particular they showed that the EpiSC state does not represent a common intermediate between ES cells and germ layers specification in vitro but an intermediate state of endodermal differentiation. Moreover, by using motif activity computational analysis they defined a transcriptional network governing mouse ES cells differentiation and the essential nodes which were validated by using CRISPR/Cas9 knockouts in a multicolor fluorescence reporter system. Finally they identified Sp1 as a general inhibitor of differentiation, Nr5a2 as a regulator of ectoderm differentiation and Fos:Jun and Zfp354c as switches between ectoderm and mesoderm fate. Overall the results are potentially interesting but the conclusion that EpiSC represent a locked state along the trajectory of endoderm al differentiation is at odds with previously published results of in-vivo derived EpiSCs and the general knowledge in the field (Kojima et., 2014 Cell Stem Cell).

Authors: This appears to be a misunderstanding by the reviewer. We do not claim that EpiSCs are not able to differentiate, only that they resemble an intermediate (but pluripotent) differentiation

state that is stable in culture. We clarify the issue in the answer to point 1 below.

Moreover, the generation of a fluorescent reporter line is potentially useful to monitor germ layers specification during differentiation of mouse ES cells although some experimental validations need to be performed.

In order to improve the impact of this work I advise the authors to address the following points:

1) the authors concluded that EpiSCs represent an intermediate state of endodermal differentiation. To demonstrate the correct differentiation of mES cells into EpiSCs *in vitro* they should show expression of markers of naïve and primed pluripotency in their *in-vitro* derived EpiSCs (Figure 3A). Also, in figure 3B and 3C they should include published embryo-derived EpiSCs as control in the PCA and heatmap.

Authors: Markers of naïve and *in vitro* derived EpiSCs were displayed in Supplementary Figure S3. We will take into account the reviewer's suggestion and display these markers in the main figure. We want to stress that from day 9 onwards of EpiSC differentiation, the gene expression profiles were similar indicating that we reached a stable population of EpiSCs. In addition, we will compare our data with published embryo-derived EpiSCs.

If EpiSCs remained locked at an intermediate state of differentiation, why are these cells used to differentiate towards the three germ layers in figure 4B and used to generate the gene regulatory network?

Authors: This appears to be a misunderstanding by the reviewer. EpiSCs are pluripotent cells that can form derivatives of the three germ layers (a fact well established in the literature -see the references cited in the text, Brons et al., 2007 and Tesar et al., 2007). We meant to say with the term "locked" that EpiSCs can be maintained stably in such a state if provided suitable signaling conditions that support self-renewal in this state. Once differentiation cues are applied, EpiSCs differentiate, as our data abundantly demonstrates. Similarly, mESCs are 'stabilized' in a naïve state if they are kept under ground-state conditions. This point is just a question of wording, really. We will use "stabilized" instead.

2) the authors generated a triple knock-in reporter line to monitor germ layer specification. However, they did not show specific up regulation of the three germ layer markers upon differentiation. First, the authors should show the basal activity of the 3 reporters in mouse ES cells under LIF+Serum conditions. It is currently hard to tell if the activity reported in figure 5 is basal/background or specific. Thus, in figure 5B-C-D fluorescence profiles of pluripotent mES should also be included.

Authors: We will provide the fluorescent profiles of pluripotent mESCs. We would like to point out that both Sox1 and Brachyury are established markers and were demonstrated to have no expression in pluripotent mESCs (see for example the two references cited in the text: Ying et al., 2003, and Fehling et al., 2003).

Second, to confirm the specificity of their reporters, the authors should sort positive and negative reporter cells and show specific upregulation of mRNA of the relative marker (eg. GFP positive cells during ectoderm differentiation should show high levels of Sox1 as compared to GFP negative cells and undifferentiated cells).

Authors: We will provide such data.

Finally, spontaneous differentiation is used in figure 5E, 5F and 6E-F. How is this spontaneous differentiation achieved? Which culture conditions?

Authors: We will add the following description to the methods: "mESCs were seeded at a density of 10,000-20,000 cells per cm² onto 0.1% gelatin coated dishes one day prior to the start of the

differentiation procedure. The following day, cells were rinsed in D-PBS and switched to Advanced RPMI 1640 (ThermoFisher) supplemented with 1 μ M IDE-1 (Tocris), 0.2% (v/v) fetal calf serum (Millipore), 2 mM L-glutamine (Sigma). Medium was replaced every day."

3) the KO generated by CRISPR/Cas9 have been validated by genomic PCR followed by sequencing, but the results presented are not clear. In 5 out of 8 lines the authors reported the presence of the same mutation on both alleles (indicated as "edited alleles"). This is very unlikely, as the most common outcome is the generation of two different mutant alleles. The authors should explain / present their results more clearly and an independent validation of their KO lines (by Western Blot / Immunostaining) would be preferable.

Authors: Explaining how cells repair CRISPR/Cas9 lesions differently for different genomic locations seems to us beyond the scope of our study. We confidently recover the wild type allele and the edited allele in the case of heterozygous lines. We offer to validate the KO lines by Western Blot analysis for the genes with an existing commercial antibody validated in mouse cells.

4) The authors propose, based on their reporter assays, that Nr5a2, Fos:Jun and Zfp354c are critical regulators of germ layer formation. Such claims are based only on reporter assay where 3 genes are monitored at a single point in time. For at least one of the genes identified it would be important to show that differentiation is actually impaired. For example, do Nr5a2 KO cells form TUJ1+ cells when exposed to Neuroectoderm differentiation protocol?

Authors: We will perform the suggested experiment to see if Nr5a2 KO cells form TUJ1+ cells.

5) Minor points: In order to better dissect gene expression changes of mESC differentiation towards the three germ layers the authors should generate 2 venn diagrams plotting separately up and down regulated genes during differentiation of the 3 germ layers (figure 1C). The same analysis should be performed for heatmaps in figure 1D-E-F.

Authors: We are happy to provide the suggested Venn diagrams.

The hierarchical clustering in Figure2A show that already from day1 ectodermal cells clustered with differentiated tissues. Such a divergence is not clear in PCA analysis in Figure 2C.

Authors: The PCA analysis in Figure 2C displays the projection of expression profiles on two dimensions explaining 65% of the variation in gene expression across samples differentiated to the three germ layers. It should be noted that day ectodermal cells are further advanced along principal component 1 than day 2 endodermal cells and day 3 mesodermal cells.

Typos / mistakes are listed below:

Page1, first paragraph:

figure 1B is Figure 1A.

Figure 1C is Figure 1B

Page2, second paragraph:

Figure2E is mentioned but it is not present in the figures.

Page7, second paragraph, second line:

Is endoderm or mesoderm differentiation described?

Page7, second paragraph, fourth line:

"differentiation under endoderm promoting conditions" but in the figure legend is "spontaneous differentiation"

Page12, third paragraph:

"maximal expression >6" should be changed into "minimal expression >6"

Authors: We thank the reviewer for pointing out these typos/mistakes and we will correct them.

2nd Editorial Decision

2nd August 2019

Thank you for your message asking us to reconsider our decision regarding your manuscript MSB-19-9043. I have carefully read your manuscript and the referee reports once again and have also discussed your preliminary point-by-point response with my colleagues. Based on the outline you provide, we think that the proposed revisions sound reasonable. As such, we would not be opposed to considering a revised version of your study.

Some of the key issues that would need to be addressed are the following:

- The main findings need to be placed in the context of previous literature in order to clarify their novelty and significance.
- The *in vivo* relevance of the main findings needs to be better supported.
- The differentiation state of EpiSCs needs to be better supported by experimental evidence and precisely described, in order to avoid misunderstandings by the reviewers.
- Further control analyses need to be included as suggested by reviewer #3.

All other issues raised by the reviewers would need to be convincingly addressed. Please include in your submission a point-by-point response to the points raised by the reviewers.

Please note that the revised version should be submitted ****within three months****. Revised manuscripts submitted after the deadline are editorially evaluated afresh and the novelty is re-assessed at the time of submission. If you anticipate that the revisions will take longer than three months, please contact me to discuss a potential extension of the deadline.

2nd Revision - authors' response

8th October 2019

Reviewer #2:

The differentiation of mouse ES cells in adherent culture has been the subject of intense scrutiny for the last ten years so, any piece of work attempting to make a significant contribution has a very high bar to rise to. In this report, Sladitschek and Neveau claim to make such a contribution. Unfortunately the report, while containing a fair amount of work and information, it falls very short of the mark. The main problem is that the culture conditions are ill defined and do not reflect the normal differentiation of the cells *in vivo* but even if the conditions were right, there is an ample literature on the subject that the authors ignore (some examples below) and which should have been discussed to provide evidence for novelty and insight.

Authors: We thank the reviewer for recognizing the large amount of information present in our manuscript We provide a detailed response to the comments that “the culture conditions are ill defined” and “there is an ample literature on the subject that the authors ignore” when the reviewer raises these points again below.

Furthermore, it is surprising that with the recent flurry of single cell papers on the early postimplantation development of the mouse, the authors do not make any attempt to check their results against them.

Authors: We have compared our findings with spatially resolved transcriptomes of gastrulating mouse embryos from Peng et al (Nature, 2019). This new analysis is shown in Figure EV1. We find that the first steps of endoderm differentiation recapitulate primitive streak formation (from which definitive endoderm arises) in the mouse embryo while *in vitro* mesoderm differentiation resembles the proximal embryonic mesoderm formation and *in vitro* ectoderm differentiation resembles ectoderm specification from the anterior epiblast. Thus *in vitro* fate acquisition mirrors fate specification that occurs in the embryo. Moreover, this analysis supports one of our key findings that EpiSCs (which resemble primitive streak cells) do not constitute an obligatory differentiation step.

There is also much that is misleading, in particular their interpretation of EpiSCs and the experiments they do with them. Overall, while I appreciate the work and care that has gone into this work, there is a naivete about the field and an ignorance of the literature that cannot be ignored.

Authors: We find this comment particularly unfair as it relies both on a gross misrepresentation of our findings and on the reviewer overlooking data presented in the manuscript (as detailed below).

In terms of the protocols, the authors use old and ill defined conditions; there are plenty of reports which they should look at that use better conditions closer to the embryo one see e.g Loh et al. (2014) Efficient endoderm induction from human pluripotent stem cells by logically directing signals controlling lineage bifurcations PMID 24412311 and if they go to PubMed, they can find many related references.

Authors: Our aim is to study how mESCs respond to differentiation cues (using well established and widely used protocols that yield specific and functional cell types) and the relationship between the acquisition of different fates. The criticism that the conditions are “ill-defined” is difficult to understand. Notably, the ectoderm and mesoderm differentiations use chemically defined media and the endoderm differentiation is chemically induced. The endoderm differentiation condition we use in the manuscript has been shown to generate functional definitive endoderm by Borowiak et al. (2009). Finally, we would like to point that the differentiating cells go through a primitive streak-like stage under the endoderm differentiation conditions we use, recapitulating the specification of definitive endoderm in the embryo. In fact, the publication taken as an example by the reviewer (PMID 24412311) follows the same differentiation path.

What is also concerning is statements like the expression of Gata 6 six days into differentiation is a reflection of endoderm differentiation when, actually, at that time (and the pathways of *in vitro* adherent differentiation are very well outlined) is a marker of cardiac differentiation.

Authors: Gata6 is an established endoderm marker that is reported to be upregulated (along with other endoderm markers) by Borowiak et al. (2009) under the endoderm differentiation conditions they pioneered, the *very same* conditions we use in this manuscript. In addition, Borowiak et al. (2009) report that “Endoderm induction from mouse ESCs peaks at day 6 of treatment with small molecules”. In the absence of any specific literature reference, the assertion of the reviewer that Gata6 “is a marker of cardiac differentiation” must therefore apply to another experimental system.

If they have any trace of endoderm, they should check in a more rigorous manner.

Authors: The reviewer completely ignores the fact that we carried out time-course RNA-Seq for the three differentiation protocols (among which endoderm differentiation), the immunostainings displayed in Figure 1A being for illustration purposes. We showed a time course of expression of endoderm markers in Figure EV2 (previously Figure S3).

Their apparently surprising result that EpiSCs can only be differentiated into endoderm (under their experimental conditions) is easily explained by the realization that they culture the cells in Activin and FGF which will direct the cells into endoderm. EpiSCs can be differentiated into the three germ layers (the literature is ample but here two examples: Kojima et al. (2014) the transcriptional and functional properties of mouse epiblast stem cells resemble anterior primitive streak PMID 24139757 and Edri et al. (2019) An epiblast stem population derived multipotent population for axial extension PMID 31023877

Authors: The reviewer’s statement “Their apparently surprising result that EpiSCs can only be differentiated into endoderm” is surprising as we never stated such a thing. In fact, we differentiated EpiSCs to the three germ layers (Figure 4B and C). In addition, we wrote page 6 (beginning of second paragraph): “While we found that *in vitro* differentiation protocols allow to bypass primed pluripotency, EpiSCs are themselves pluripotent”. To prevent any misunderstanding, we have now introduced a Figure 7 in which we summarize our main findings.

The network analysis is fine, though neither novel nor insightful but it is well done; unfortunately given the problems with the data, the output reflects the response of the cells to the culture conditions rather than what actual happens *in vivo*. The genetic loss of function experiments follow suit. The basic idea is good and the reporter and strategy are all fine but the culture conditions are very artificial and we all know that cells will respond to cocktails of signals in interesting manners.

Authors: The aim of our study is precisely to understand how cells respond to the different differentiation conditions (which are published and were shown to create functional cell types). Understanding mESC differentiation is of fundamental importance for regenerative medicine applications but insights gained from such studies might also be applicable to embryonic development. While we revised this manuscript, Sp1 was recently found as an important regulator of fate specification in the gastrulating mouse embryo from analysis of spatially defined transcriptomes (Peng et al., Nature, 2019). These authors' Sp1^{-/-} mESCs have the same phenotype of facilitating exit of naïve pluripotency as our Sp1^{+/-} mESCs. Finally, *in vitro* fate acquisition mirrors fate specifications that occurs in the embryo as demonstrated by the comparison of our data with the *in vivo* data of Peng et al (Nature, 2019).

Reviewer #3:

The manuscript of Sladitschek and Neveu describes the identification of regulators of mouse ES cells differentiation.

In particular they showed that the EpiSC state does not represent a common intermediate between ES cells and germ layers specification *in vitro* but an intermediate state of endodermal differentiation. Moreover, by using motif activity computational analysis they defined a transcriptional network governing mouse ES cells differentiation and the essential nodes which were validated by using CRISPR/Cas9 knockouts in a multicolor fluorescence reporter system. Finally they identified Sp1 as a general inhibitor of differentiation, Nr5a2 as a regulator of ectoderm differentiation and Fos:Jun and Zfp354c as switches between ectoderm and mesoderm fate.

Overall the results are potentially interesting but the conclusion that EpiSC represent a locked state along the trajectory of endodermal differentiation is at odds with previously published results of *in vivo* derived EpiSCs and the general knowledge in the field (Kojima et., 2014 Cell Stem Cell).

Authors: This appears to be a misunderstanding by the reviewer. We do not claim that EpiSCs are not able to differentiate, only that they resemble an intermediate (but pluripotent) differentiation state that is stable in culture. We clarify the issue in the answer to point 1 below. To prevent any misunderstanding, we have now introduced a Figure 7A in which we illustrate the hierarchy of naïve and primed pluripotency and germ layer acquisition.

Moreover, the generation of a fluorescent reporter line is potentially useful to monitor germ layers specification during differentiation of mouse ES cells although some experimental validations need to be performed.

In order to improve the impact of this work I advise the authors to address the following points:

1) the authors concluded that EpiSCs represent an intermediate state of endodermal differentiation. To demonstrate the correct differentiation of mES cells into EpiSCs *in vitro* they should show expression of markers of naïve and primed pluripotency in their *in-vitro* derived EpiSCs (Figure 3A). Also, In figure 3B and 3C they should include published embryo-derived EpiSCs as control in the PCA and heatmap.

Authors: Some markers of naïve and *in vitro* derived EpiSCs were displayed in Supplementary Figure S3. Following the reviewer's suggestion, we have compared our *in vitro* derived EpiSCs with embryo-derived EpiSCs from Tesar et al. (2007). Expression levels of naïve and primed pluripotency markers are shown in Figure EV2C and D and the expression profiles of embryo-derived EpiSCs project like our *in vitro* derived EpiSCs on the PCA map (shown in Figure EV2E and F).

If EpiSCs remained locked at an intermediate state of differentiation, why are these cells used to differentiate towards the three germ layers in figure 4B and used to generate the gene regulatory network?

Authors: This appears to be a misunderstanding by the reviewer. EpiSCs are pluripotent cells that can form derivatives of the three germ layers (a fact well established in the literature –see the references cited in the text, Brons et al., 2007 and Tesar et al., 2007). We meant to say with

the term “locked” that EpiSCs can be maintained stably in such a state if provided suitable signaling conditions that support self-renewal in this state. Once differentiation cues are applied, EpiSCs differentiate, as our data abundantly demonstrates. Similarly, mESCs are ‘*stabilized*’ in a naïve state if they are kept under ground-state conditions. This point is just a question of wording, really. We will use “stabilized” instead. Moreover, to prevent any misunderstanding, we have now introduced a Figure 7A in which we illustrate the hierarchy of naïve and primed pluripotency and germ layer acquisition.

2) the authors generated a triple knock-in reporter line to monitor germ layer specification. However, they did not show specific up regulation of the three germ layer markers upon differentiation. First, the authors should show the basal activity of the 3 reporters in mouse ES cells under LIF+Serum conditions. It is currently hard to tell if the activity reported in figure 5 is basal/background or specific. Thus, in figure 5B-C-D fluorescence profiles of pluripotent mES should also be included.

Authors: We now provide the fluorescent profiles of pluripotent mESCs in Figure EV4C, E and G showing the absence of reporter expression in undifferentiated mESCs. In addition, we would like to point out that both *Sox1* and *Brachyury* are established markers and were demonstrated to have no expression in pluripotent mESCs (see for example the two references cited in the text: Ying et al., 2003 using the exact same *Sox1*-GFP line that is the parental line for our triple knock-in, and Fehling et al., 2003).

Second, to confirm the specificity of their reporters, the authors should sort positive and negative reporter cells and show specific upregulation of mRNA of the relative marker (eg. GFP positive cells during ectoderm differentiation should show high levels of *Sox1* as compared to GFP negative cells and undifferentiated cells).

Authors: We FACS-purified differentiating GFP-positive and GFP-negative cells and measured *Sox1* and GFP mRNA levels as well as TagBFP-positive and TagBFP-negative cells and measured T and TagBFP mRNA levels. This new data confirms the specificity of the reporters and is displayed in Figure EV4D and F.

Finally, spontaneous differentiation is used in figure 5E, 5F and 6E-F. How is this spontaneous differentiation achieved? Which culture conditions?

Authors: We added the following description to the methods: “For spontaneous differentiation between mesendoderm and ectoderm fates, mESCs were seeded at a density of 10,000-20,000 cells per cm² onto 0.1% gelatin coated dishes one day prior to the start of the differentiation procedure. The following day, cells were rinsed in D-PBS and switched to Advanced RPMI 1640 (ThermoFisher) supplemented with 1 µM IDE-1 (Tocris), 0.2% (v/v) fetal calf serum (Millipore), 2 mM L-glutamine (Sigma). Medium was replaced every day.”

3) the KO generated by CRISPR/Cas9 have been validated by genomic PCR followed by sequencing, but the results presented are not clear. In 5 out of 8 lines the authors reported the presence of the same mutation on both alleles (indicated as “edited alleles”). This is very unlikely, as the most common outcome is the generation of two different mutant alleles. The authors should explain / present their results more clearly and an independent validation of their KO lines (by Western Blot / Immunostaining) would be preferable.

Authors: We validated all the KO lines by Western Blot analysis with commercial antibody validated in mouse cells (data displayed in Figure EV5). Explaining how cells repair CRISPR/Cas9 lesions differently for different genomic locations seems to us beyond the scope of our study. We confidently recover the wild type allele and the edited allele in the case of heterozygous lines.

4) The authors propose, based on their reporter assays, that *Nr5a2*, *Fos:Jun* and *Zfp354c* are critical regulators of germ layer formation. Such claims are based only on reporter assay where 3 genes are monitored at a single point in time. For at least one of the genes identified it would be important to show that differentiation is actually impaired. For example, do *Nr5a2* KO cells form TUJ1+ cells when exposed to Neuroectoderm differentiation protocol?

Authors: We performed the suggested experiment to see if Nr5a2 KO cells can generate TUJ1+ cells. The formation of TUJ1-positive cells upon neuroectoderm differentiation of Nr5a2 KO cells is indeed greatly impaired (Appendix Figure S3).

5) Minor points: In order to better dissect gene expression changes of mESC differentiation towards the three germ layers the authors should generate 2 venn diagrams plotting separately up and down regulated genes during differentiation of the 3 germ layers (figure 1C). The same analysis should be performed for heatmaps in figure 1D-E-F.

Authors: We now provide the suggested Venn diagrams for genes upregulated during differentiation or downregulated during differentiation in Figure 1C. The heatmaps in figure 1D-E-F display a single condition and it is thus impossible to generate a Venn diagram (that necessitates the comparison of at least two conditions).

The hierarchical clustering in Figure2A show that already from day1 ectodermal cells clustered with differentiated tissues. Such a divergence is not clear in PCA analysis in Figure 2C.

Authors: The PCA analysis in Figure 2C displays the projection of expression profiles on two dimensions explaining 65% of the variation in gene expression across samples differentiated to the three germ layers. It should be noted that day 1 ectodermal cells are further advanced along principal component 1 than day 2 endodermal cells and day 3 mesodermal cells.

Typos / mistakes are listed below:

Page1, first paragraph:

figure 1B is Figure 1A.

Figure 1C is Figure 1B

Authors: We thank the reviewer for pointing out these typos/mistakes and we have corrected them.

Page2, second paragraph:

Figure2E is mentioned but it is not present in the figures.

Authors: We thank the reviewer for noticing this reference to a previous version of the figure. We deleted that reference in the new version.

Page7, second paragraph, second line:

Is endoderm or mesoderm differentiation described?

Authors: Mesoderm differentiation was described and the text has been changed accordingly.

Page7, second paragraph, fourth line:

"differentiation under endoderm promoting conditions" but in the figure legend is "spontaneous differentiation"

Authors: We modified the sentence to refer to the Materials and Methods in which we describe in detail the differentiation conditions enabling the simultaneous formation of GFP-positive cells, TagBFP-positive cells and mCherry-positive cells. We refer to these conditions as "spontaneous differentiation between mesendoderm and ectoderm fates".

Page12, third paragraph:

"maximal expression >6" should be changed into "minimal expression >6"

Authors: We kept genes for the analysis with maximal expression >6 RPM not to filter out genes that are upregulated or downregulated during differentiation (for example pluripotency factors that have low expression in differentiated cells or conversely transcription factors only expressed in differentiated cells).

Thank you for sending us your revised manuscript. We have now heard back from the two reviewers who agreed to evaluate your manuscript. You will see from the comments below that both reviewer #3 (who had reviewed the manuscript before) and reviewer #4 (who reviewed the manuscript for the first time) are overall positive and support publication of the article in *Molecular Systems Biology*. However, the reviewers also raised a couple of minor issues that may still be addressed to avoid potential confusion on the readers. We would encourage you to address them within reason.

REFEREE REPORTS

Reviewer #3:

The authors addressed in a satisfactory way all points that were raised, therefore we feel that the current version of the manuscript is suitable for publication.

There is only one point that should be clarified, given that it caused confusion on both reviewers and it may cause confusion also on the readers.

The authors claim that EpiSCs occupy a stable intermediate state along the Endoderm differentiation trajectory and therefore the Primed state of pluripotency "does not constitute a obligatory intermediate differentiation state".

It should be made very clear that EpiSCs are pluripotent, despite the expression of some markers of germ layers. Indeed, since their initial derivation (Tesar et al., 2007), the markers *Foxa2*, *Gata6*, *Sox17*, *Cer1* have been identified as upregulated genes in EpiSCs compared to ES cells. Such genes are potently induced by TGF-beta ligands or activators (such as Activin or IDE-1), that are used for both the expansion of EpiSCs or for Endoderm differentiation. So, the transcriptional similarities between EpiSCs and cells differentiating towards Endoderm might simply reflect a response to the signaling environment, with no functional relevance. Indeed, EpiSCs are pluripotent, and not committed toward Endoderm differentiation, so such transcriptional similarities do not reflect any bias in differentiation.

As both reviewers pointed out, such findings should be discussed in the context of the literature, where the similarity between EpiSCs and the primitive streak was already reported (Kojima et al, 2014). Based on the known similarity between EpiSCs and primitive streak, it is not surprising that ES cells differentiation towards neurectoderm do not go through an "EpiSC state".

In sum, the authors should tone down, or remove, the claims concerning differentiation trajectories and primed state, that are marginal to their conclusions. It would be fair to say that their results agree with previous observations, reporting similarities between EpiSCs and primitive streak, and that such transcriptional similarities might be the product of in vitro culture in presence of TGF-beta, with no functional relevance, given that pluripotency is maintained despite the expression of some markers of germ layers.

The second part of the manuscript (network analysis, identification and functional testing of key TFs) is very well done and deserves to be at the readers' centre of the attention.

Reviewer #4:

Sladitschek and Neveu use bulk RNA-seq in an in vitro differentiation system of mES to the three germ layers. They identify several potential regulators of these different differentiation paths and use CRISPR deletions to confirm their role in differentiation. In addition they also present evidence that primed pluripotency represented by EpiSCs is not an obligatory intermediate state for all three lineages.

Overall I find the work sufficiently novel and well executed (especially after addressing the issues arising in revision) to merit publication in MSB. The major issues that have been mostly addressed in revision are:

1. How well-defined are the culture conditions used and how similar to their *in vivo* counterparts are the resulting cells?

The culture conditions that the authors used are well-established, lead to a highly homogenous populations of targeted cells and have been previously used in many studies. Of course, they do not completely recapitulate embryogenesis *in vivo* but the authors have mostly refrained from making such exaggerated claims. Such experiments however, are well-suited to understand the molecular mechanisms of how cell fate decisions occur during differentiation. As response to the review 1 comments, the authors compare the differentiation trajectory with spatial transcriptomes from gastrulating embryos. Although not identical, *in vitro* cell fate acquisition is sufficiently similar to the process *in vivo*.

2. How restricted are EpiSCs and are they a necessary intermediate state for all germ layers?

A major issue during revision was the apparent statement that EpiSCs represent a "locked" state during endoderm differentiation. According to the author's response, this was not the intended conclusion and they have provided evidence that EpiSCs can indeed differentiate into all three lineages as expected. They have also introduced a summary figure to try to avoid misinterpretation. While reading the revised text, I did not get the impression that the authors implied the EpiSCs cannot differentiate into other layers. However, to avoid any misinterpretations it would be good if the authors can show the EpiSCs directly on their global differentiation PCA (that is project the EpiSCs on the plot in Figure 2C) - this can be included in one of the supplementary figures and stated in the text. They also compare their EpiSCs with embryo-derived and show that they are very similar transcriptionally.

I find the rest of the responses to the previous reviewer comments satisfactory.

3rd Revision - authors' response

11th November 2019

Reviewer #3:

The authors addressed in a satisfactory way all points that were raised, therefore we feel that the current version of the manuscript is suitable for publication.

Authors: We thank the reviewer for her/his positive evaluation of our work.

There is only one point that should be clarified, given that it caused confusion on both reviewers and it may cause confusion also on the readers.

The authors claim that EpiSCs occupy a stable intermediate state along the Endoderm differentiation trajectory and therefore the Primed state of pluripotency "does not constitute a obligatory intermediate differentiation state".

It should be made very clear that EpiSCs are pluripotent, despite the expression of some markers of germ layers. Indeed, since their initial derivation (Tesar et al., 2007), the markers *Foxa2*, *Gata6*, *Sox17*, *Cer1* have been identified as upregulated genes in EpiSCs compared to ES cells. Such genes are potentially induced by TGF-beta ligands or activators (such as Activin or IDE-1), that are used for both the expansion of EpiSCs or for Endoderm differentiation. So, the transcriptional similarities between EpiSCs and cells differentiating towards Endoderm might simply reflect a response to the signaling environment, with no functional relevance. Indeed, EpiSCs are pluripotent, and not committed toward Endoderm differentiation, so such transcriptional similarities do not reflect any bias in differentiation.

As both reviewers pointed out, such findings should be discussed in the context of the literature, where the similarity between EpiSCs and the primitive streak was already reported (Kojima et al,

2014). Based on the known similarity between EpiSCs and primitive streak, it is not surprising that ES cells differentiation towards neurectoderm do not go through an "EpiSC state".

In sum, the authors should tone down, or remove, the claims concerning differentiation trajectories and primed state, that are marginal to their conclusions. It would be fair to say that their results agree with previous observations, reporting similarities between EpiSCs and primitive streak, and that such transcriptional similarities might be the product of in vitro culture in presence of TGF-beta, with no functional relevance, given that pluripotency is maintained despite the expression of some markers of germ layers.

Authors: We agree with the reviewer that the signaling environment could be a determining factor explaining the similarities between EpiSCs and endoderm differentiation. We had mentioned such a possible explanation as well as the similarities between EpiSCs and the primitive streak in the discussion, citing indeed the study by Kojima et al. pointed out by the reviewer. Following the reviewer's recommendation, we now stress that EpiSCs are pluripotent. In addition to dispel the potential misunderstanding that EpiSCs might be committed to an endoderm fate, we added the following sentence in the discussion: "It should be noted that EpiSCs are pluripotent and that their differentiation potential is therefore not restricted by the expression of germ layer markers". While it might be "not surprising that ES cells differentiation towards neurectoderm do not go through an "EpiSC state"" according to the reviewer, such a finding had not been reported previously. The comparison of differentiation trajectories is important to understand how embryonic stem cells acquire different fates, an opinion shared by Reviewer #4.

The second part of the manuscript (network analysis, identification and functional testing of key TFs) is very well done and deserves to be at the readers' centre of the attention.

Authors: We thank the reviewer for her/his positive evaluation of our work.

Reviewer #4:

Sladitschek and Neveu use bulk RNA-seq in an in vitro differentiation system of mES to the three germ layers. They identify several potential regulators of these different differentiation paths and use CRISPR deletions to confirm their role in differentiation. In addition they also present evidence that primed pluripotency represented by EpiSCs is not an obligatory intermediate state for all three lineages.

Overall I find the work sufficiently novel and well executed (especially after addressing the issues arising in revision) to merit publication in MSB.

Authors: We thank the reviewer for her/his positive evaluation of our work.

The major issues that have been mostly addressed in revision are:

1. How well-defined are the culture conditions used and how similar to their in vivo counterparts are the resulting cells?

The culture conditions that the authors used are well-established, lead to a highly homogenous populations of targeted cells and have been previously used in many studies. Of course, they do not completely recapitulate embryogenesis in vivo but the authors have mostly refrained from making such exaggerated claims. Such experiments however, are well-suited to understand the molecular mechanisms of how cell fate decisions occur during differentiation. As response to the review 1 comments, the authors compare the differentiation trajectory with spatial transcriptomes from gastrulating embryos. Although not identical, in vitro cell fate acquisition is sufficiently similar to the process in vivo.

Authors: We thank the reviewer for her/his supportive evaluation of the revision of our work.

2. How restricted are EpiSCs and are they a necessary intermediate state for all germ layers?

A major issue during revision was the apparent statement that EpiSCs represent a "locked" state during endoderm differentiation. According to the author's response, this was not the intended conclusion and they have provided evidence that EpiSCs can indeed differentiate into all three lineages as expected. They have also introduced a summary figure to try to avoid misinterpretation. While reading the revised text, I did not get the impression that the authors implied the EpiSCs cannot differentiate into other layers. However, to avoid any misinterpretations it would be good if the authors can show the EpiSCs directly on their global differentiation PCA (that is project the EpiSCs on the plot in Figure 2C) - this can be included in one of the supplementary figures and stated in the text. They also compare their EpiSCs with embryo-derived and show that they are very similar transcriptionally.

Authors: We thank the reviewer for her/his positive evaluation the revision of our manuscript. The projection of EpiSC profiles on the global differentiation PCA is displayed in Figure 3B and Figure EV2E and EV2F while the trajectory of EpiSC derivation from mouse Embryonic Stem Cells is shown in Figure 3B.

I find the rest of the responses to the previous reviewer comments satisfactory.

Authors: We thank the reviewer for her/his positive evaluation of our work.

Accepted

14th November 2019

Thank you again for sending us your revised manuscript. We are now satisfied with the modifications made and I am pleased to inform you that your paper has been accepted for publication.

Corresponding Author Name: Pierre A. Neveu
Journal Submitted to: Molecular Systems Biology
Manuscript Number: MSB-19-9043